# A Modified Variant of *Fasciola hepatica* FhSAP-2 (mFhSAP-2) as a Recombinant Vaccine Candidate Induces High-Avidity IgG2c Antibodies and Enhances T Cell Activation in C57BL/6 Mice

**DOI:** 10.3390/vaccines13050545

**Published:** 2025-05-20

**Authors:** Riseilly Ramos-Nieves, Albersy Armina-Rodriguez, Maria Del Mar Figueroa-Gispert, Ghalib Figueroa-Quiñones, Carlimar Ocasio-Malavé, Ana M. Espino

**Affiliations:** 1Department of Microbiology and Medical Zoology, School of Medicine, University of Puerto Rico, Medical Sciences Campus, San Juan, PR 00936, USA; riseilly.ramos@upr.edu (R.R.-N.); albersy.armina@upr.edu (A.A.-R.); maria.figueroa14@upr.edu (M.D.M.F.-G.); carlimar.ocasio@upr.edu (C.O.-M.); 2Department of Biology, University of Puerto Rico, Arecibo Campus, Arecibo, PR 00614, USA; ghalib.figueroa@upr.edu

**Keywords:** C57BL6, IgG2a, IgG2c, avidity, FhSAP-2, IFNγ

## Abstract

Background/Objectives: In the past, FhSAP-2, an 11.5 kDa recombinant protein belonging to the *Fasciola hepatica* saposin-like/NK-lysin family, has been shown to induce over 60% partial protection in immunized rabbits and mice when challenged with *F. hepatica* metacercariae. However, despite FhSAP-2 being a promising vaccine candidate, its hydrophobic nature has made its purification a challenging process. The present study aimed to determine whether a modified 9.8 kDa variant of protein (mFhSAP-2), lacking a string of 16 hydrophobic amino acids at the amino terminus and a dominant Th1 epitope, could retain its immunogenic and Th1-inducing properties. Methods: RAW264.7 cells were stimulated with mFhSAP-2, and TNFα levels were determined. C57BL/6 mice were immunized with mFhSAP-2 alone or emulsified with Montanide ISA50. Total anti-mFhSAP-2 IgG subtypes, along with their avidity and titers, were measured using ELISA. The T cell proliferation index and levels of CD4+/CD8+ and IFNγ/IL-4 ratios were determined. Results: In vitro, mFhSAP-2 induced dose-dependent TNFα production in RAW264.7 cells. In vivo, mice immunized with mFhSAP-2 or mFhSAP-2+ISA50 developed high-avidity IgG2a and IgG2c antibodies at levels that were significantly higher than IgG1 antibody levels. However, the mFhSAP-2+ISA50 formulation induced higher and more homogenous antibody titers than mFhSAP-2, suggesting that an adjuvant may be required to enhance mFhSAP-2 immunogenicity. Immunization with mFhSAP-2+ISA50 also induced significantly higher activated CD4+/CD8+ T cell ratios and IFNγ/IL-4 ratios compared to naïve mice. Conclusions: Our results demonstrate that mFhSAP-2 retained its immunogenicity and Th1-polarizing properties, which were enhanced by the Montanide ISA50 adjuvant. The present study highlights the feasibility of inducing Th1-associated immune responses in mice using mFhSAP-2 as an antigen. Further studies are required to assess the potential application of the mFhSAP-2+ISA50 formulation as a vaccine against *F. hepatica* in natural hosts such as cattle and sheep, which could contribute to improved control and aid in the prevention and eradication of *F. hepatica* infection.

## 1. Introduction

Fasciolosis, caused by *Fasciola hepatica*, is a globally distributed zoonotic disease that causes significant economic losses to agricultural and husbandry industries worldwide, which have been estimated to be more than USD 3.2 billion/year [1,2]. Fasciolosis also affects over 2.4 million people worldwide and has been recently declared an emerging neglected tropical disease by the World Health Organization (WHO) [3,4]. The drug of choice for treating fasciolosis is triclabendazole. However, the emergence of resistance to this drug in diverse *F. hepatica* populations [5,6], along with the high costs associated with this treatment, suggests the need for alternative control strategies such as vaccines. Over the last three decades, there have been numerous attempts to develop a successful vaccine against *F. hepatica* using different strategies, including the use of single molecules or mixtures of antigens [2,7]. Excretory–secretory products (ESPs) represent the lead *F. hepatica* vaccine candidates, which include native and recombinant variants of fatty-acid-binding proteins (FABPs) [8,9,10], glutathion-S-transferase (GST) [11,12,13], Cathepsin-L (CatL1) [7,14,15], leucinaminopeptidase (LAP) [15,16,17], Kunitz-type inhibitors [18,19], and thioredoxin–peroxiredoxin–peroxidases [20,21], among others. These vaccines are yet to achieve substantial and reproducible protection, with a mean efficacy of ~40.0 to 76.0% in mice or rabbits [8,9] and an average protection of 34.7% in goats, sheep, and cattle [9,11,12,13,22]. The molecules of the parasite mediate a range of relevant functions for the parasite’s invasion, migration, nutrition, and survival in the host. Therefore, an effective vaccine should aim to block some of the parasites’ functions, which are highly expressed throughout their juvenile and adult stages. Interestingly, some of the most popular vaccine candidates (ESPs, FABPs, GST, and CatL1) are also considered strong immunomodulatory molecules that, along with a myriad of other parasite molecules, can interact with cells of the immune system to inhibit Th1-driven responses through the induction of alternative activated macrophages [23] and mast cells [24] and the establishment of Th2/Treg immune responses [25,26,27]. Since Th2/Treg immune responses are ineffective in eliminating the parasite, allowing its establishment in the host, and the development of chronic fasciolosis and since previous studies have revealed that animals that exhibit a natural resistance to *Fasciola gigantica* infection show significantly low IL-4/IFNγ RNA expression and a specific low IgG1/IgG2 antibody response [28], it is possible to hypothesize that Th1 immune responses could also be required to induce protective immunity against *F. hepatica*. Our research group has focused on identifying *F. hepatica* molecules that are expressed from the early stages of infection and have the ability to induce Th1 immune responses. This approach aims to limit the fluke-induced Th2 response, which is imperative for developing an effective vaccine against liver fluke infection.

FhSAP-2 is an ~11.5 kDa recombinant polypeptide that belongs to the *F. hepatica* saposin-like/NK-lysin protein family [29]. It is highly expressed during several stages of a parasite’s life cycle, including undeveloped eggs, newly excysted juveniles (NEJs), and adult flukes [30]. FhSAP-2 also contains two dominant B cells and Th1 epitopes in its protein moiety [31,32]. When FhSAP-2 was administered subcutaneously (SC) in complete Freund’s adjuvant (CFA) or in the form of inclusion bodies (IBs), this resulted in a significant clearance of infection, ranging from 60 to 83% [33,34]. Since it is generally accepted that a liver fluke vaccine will have 60–70% efficacy in cattle [35,36] and the protection induced by FhSAP-2 in both mice and rabbits exceeds this threshold, it is highly recommended to further assess its capacity to induce protective immune responses in ruminants. However, these efforts have been largely hampered by the challenge of obtaining sufficient amounts of FhSAP-2 with a high degree of purity to scale-up vaccination trials in sheep or cattle.

FhSAP-2 is rich in hydrophobic amino acids (39.6%) [29], and this trait makes the protein very difficult to purify even when extracted from IBs due to its tendency to aggregate and form non-specific interactions with other molecules in solution. The first 16 amino acids at the amino terminus of the FhSAP-2 protein sequence comprise a string of hydrophobic and hydroxylic residues that are predicted to constitute an N-terminal signal cleavage site between amino acids Ala15 and Ser16 [29]. Therefore, a feasible strategy to enhance its solubility would be to remove this region and replace it with one that would help increase its solubility. However, considering that one of the dominant Th1 epitopes of FhSAP-2 (^11^AVTFA^15^) was experimentally localized at the end of these 16 amino acids [31], the removal of this sequence could be detrimental to its ability to induce strong Th1 immune responses, potentially affecting its protective efficacy. This study aimed to determine whether a modified version of FhSAP-2, lacking the first 16 amino acids containing a string of hydrophobic residues at the amino terminus and, consequently, missing a dominant Th1 epitope, could be easier to purify while still retaining immunogenicity comparable to the full-length protein. Additionally, this study also aimed to determine whether the modified protein (mFhSAP-2), administered alone or emulsified in a commercially well-accepted adjuvant for human or veterinary use, such as Montanide ISA50, could still induce Th1-type immune responses in mice. Since the measurement of IgG2a or IgG2c antibody subtypes, along with the production of IFNγ and IL-4 in response to a vaccine, is essential for assessing the relative contribution of Th1 versus Th2 immune responses, we compared the levels of IgG2a and IgG2c antibody subtypes, along with their avidities and titers, using ELISA. Moreover, we investigated the capacity of mFhSAP-2 to stimulate TNFα secretion from macrophage-like cells in vitro, the activation status of T cells, and their production of IFNγ and IL-4 in vivo using flow cytometry. This comprehensive approach provided a more accurate assessment of Th1/Th2 immune responses, offering deeper insight into the immune mechanisms and laying the groundwork for, in the near future, testing mFhSAP-2 as a novel vaccine candidate in sheep or cattle, which are the natural hosts of *F. hepatica*.

## 2. Materials and Methods

### 2.1. Recombinant Protein Production and Purification

The cDNA encoding full-length FhSAP-2 (GeneBank AF286903) [29] was synthesized. It comprises 306 bps and encrypts a polypeptide of 101 amino acid residues, with a calculated molecular mass of 11.5 kDa and an isoelectric point (pI) of 4.63. Because the first 48 bps of FhSAP-2 encode 16 predominantly hydrophobic amino acids (^1^MNPLFVLMLAAVTFAS^16^) and include a predicted signal peptide cleavage site between amino acids Ala15 and Ser16 [29], this sequence was omitted. Only the remaining 258 bps were synthesized and cloned into the vector pET30a (+) using the restriction enzymes Kpnl and Hind-III. Thus, the resulting polypeptide comprised 85 amino acid residues with a calculated molecular mass of 9.8 kDa, which was termed mFhSAP-2. To facilitate the detection and purification of the protein, a sequence of six histidine amino acids (6His) and a synthetic GST tag were added to the amino terminus, also enhancing its solubility. A TEV protease recognition sequence (ENLYFQS) was also inserted between the GST tag and the mFhSAP-2 coding region to allow for the recovery of the tag-free protein following enzymatic digestion (Figure 1). The fusion protein 6His-GST-mFhSAP-2 was expressed in *Escherichia coli* BL21 Star^TM^ DE3.

The expressed protein was extracted from the bacterial inclusion bodies using denaturing buffers and purified using a HiTrap Chelating HP column (Cytiva, Marlborough, MA, USA) as previously described [29]. The synthesis of cDNA, cloning, expression and purification was performed in collaboration with GenScript, USA (Order # U091CFK120). A flowchart showing a simplified sketch of the protein purification process is provided in Figure 2. After elution, the recombinant fusion protein was renatured, digested, and subjected to exhaustive cycles of endotoxin removal using polymyxin B (PMB) columns according to the manufacturer’s instructions. Endotoxin levels were assessed using a Chromogenic Limulus Amebocyte Lysate (LAL) assay (Lonza, Walkersville, MD, USA). Finally, mFhSAP-2 was concentrated using AMICON Ultracentrifugal Filters (YM-3), and its concentration was adjusted to 1.0 mg/mL, as determined by the BCA method. The purity of the protein was estimated through a densitometric analysis of a Coomassie blue-stained SDS-PAGE gel under reducing conditions. A detailed technical description of the purification process, including the buffer composition, enzyme treatments, and chromatography, is provided as Appendix A.

### 2.2. Circular Dichroism (CD)

CD spectroscopy was used to analyze the secondary structure of mFhSAP-2. The protein was diluted in PBS at pH 7.2 to achieve a concentration of ~200 µg/mL. CD spectra were collected using a J-1500 (Jasco) CD spectropolarimeter (Jasco Inc., Tokyo, Japan). Measurements were taken in a 1 mm cuvette at room temperature (25 °C) over a wavelength range of 190–250 nm. Spectra were recorded at a scan speed of 50 nm/min, with a data pitch of 0.1 nm and a bandwidth of 1 nm. Three spectra were measured to improve the signal-to-noise ratio.

The raw CD spectra data were analyzed using the BeStSel^TM^ (2014–2024) web server (https://bestsel.elte.hu/index.php) developed by ELTE Eötvös Loránd University (Budapest, Hungary) to determine the protein’s secondary structure based on experimental values. The results were then compared to a secondary structure prediction of the full-length protein generated by the SOPM server [29].

### 2.3. Animals

Wild-type inbred C57BL/6 mice (both sexes, 6–8 weeks old) were obtained from Charles River Laboratories (Wilmington, MA, USA). Animals were housed and maintained under standard conditions at 21 °C with a 12 h light–dark cycle and had ad libitum access to food and water. The animal study protocol was approved by the Institutional Animal Care and Use Committee (IACUC) of the University of Puerto Rico Medical Sciences Campus (protocols no. 7870104 and 7870106).

### 2.4. Cell Culture, mFhSAP-2 Treatment, and Measurement of TNFα Levels by ELISA

RAW264.7 murine macrophage cells were obtained from ATCC (Manassas, VA, USA). Cells were seeded at 5 × 10^5^ cells/mL in 24-well culture plates in DMEM with L-glutamine, sodium pyruvate, and sodium bicarbonate (Sigma Aldrich, St Louis, MO, USA), supplemented with 10% heat-inactivated fetal bovine serum (FBS), 100 U/mL penicillin, and 100 μg/mL streptomycin (Sigma Aldrich, USA). Cells were treated in triplicate with mFhSAP-2 at concentrations of 5, 10, 15, and 20 µg/mL and 10 µg/mL lipopolysaccharide (LPS) from *E. coli* 0111: B4 or PBS and incubated overnight at 37 °C and 5% CO_2_. After incubation, culture media were collected and centrifuged at 1000× *g* for 10 min. The supernatants were then used to measure levels of tumor necrosis factor-alpha (TNFα) using a commercially available enzyme-linked immunosorbent assay (ELISA) kit from Abcam (Cambridge, UK), following the manufacturer’s instructions.

### 2.5. Cell Viability Assay

RAW264.7 cells were seeded at 1 × 10^5^ cells/well in 96-well flat-bottom plates, treated with LPS (10 μg/mL), mFhSAP-2 (5, 10, 15, and 20 μg/mL), or PBS, and incubated for 24 h at 37 °C in 5% CO_2_. Following incubation, cell viability was assessed by adding 50 μL of XTT (sodium 3′-[1-(phenylaminocarbonyl)-3,4 tetrazolium]-bis(4-methoxy-6-nitro) benzene sulfonic acid hydrate) labeling reagent (Abcam, Cambridge, UK) to each well. After 4 h, the absorbance was read at 450 nm.

### 2.6. Sera from Fasciola hepatica-Infected Animals

The sera from animals with active *F. hepatica* infection used in this study were obtained from the repository bank of the Molecular Parasitology and Immunology Laboratory in the Department of Microbiology at the University of Puerto Rico Medical Sciences Campus. Sera were collected under IACUC protocols no. 7870104 and 7870106, as outlined in Section 2.3. Samples included sera from 16 New Zealand White (NZW) rabbits that were orally infected with 30 *F. hepatica* metacercariae (mc). Eight of these samples were collected between 3 and 6 weeks after infection (early infection), while the other eight were collected between 10 and 12 weeks after the infection (late infection). The present study also included serum samples from 11 inbred mice (6 C57BL/6 and 5 BALB/c mice) collected 15 to 21 days after oral infection with 10 *F. hepatica* mc, each suspended in tap water. The infection of both rabbits and mice was confirmed by the presence of immature or mature flukes in their respective livers [33,34]. Serum samples from naïve NZW rabbits and mice were included as negative controls (NCs). The aliquots used were kept frozen at −80 °C and had not been thawed prior to use.

### 2.7. Immunization of C57BL/6 Mice with mFhSAP-2

A group of ten C57BL/6 mice (*n* = 10) was injected subcutaneously (sc) three times with 50 μg of emulsified mFhSAP-2 in Montanide ISA50 (1:1) (Seppic, Courbevoie, France). Each immunization was administered at different sites on the dorsal surface of the mice at biweekly intervals using a sterile hypodermic G-26 needle connected to a 1 cc insulin syringe. Another group of mice (*n* = 5) received the same immunization regimen with 50 μg of mFhSAP-2 without adjuvant. Two weeks after the last immunization (day 45 after the first immunization), the animals were bled out via the retroorbital vein under deep anesthesia. A group of naïve mice (*n* = 5) that received sc injections with the adjuvant only was also bled out and used as a negative control. For blood collection, we followed the standard protocol approved by the institutional IACUC of the University of Puerto Rico Medical Sciences Campus (protocols no. 7870104 and 7870106). Briefly, a sterile hematocrit capillary tube was inserted into the medial canthus of the eye of each mouse, allowing blood to flow by capillary action into an SST^TM^ BD Container (Becton & Dickinson, NJ, USA). After gently inverting the tubes 5–10 times to mix, they were allowed to clot for 30 min at room temperature in a vertical position. Then, the SST^TM^ BD container was centrifuged at 2000 rpm for 10 min to separate the serum. The serum samples were stored in 20 μL aliquots at −80 °C until use. Animals were euthanized via cervical dislocation and necropsied for spleen collection.

### 2.8. Enzyme-Linked Immunoassay (ELISA) for Determining Total Antibody Levels of IgG and IgG Isotypes

Sera from rabbits and mice infected with *F. hepatica* and sera from mFhSAP-2-immunized mice were analyzed using ELISA to measure specific IgG antibody levels using mFhSAP-2 as the antigen, following a pre-established protocol that had been optimized by checkerboard titration [37]. After blocking with 3% skimmed milk in phosphate-buffered saline (PBS) containing 0.05% Tween 20 (PBST), each serum sample diluted 1:100 in PBST was incubated in duplicate for 1 h at 37 °C. For the detection of the total IgG antibody response, peroxidase-conjugated anti-rabbit IgG (Bio-Rad Laboratories, Hercules, CA, USA) or anti-mouse IgG (Cell Signaling Technology, Danvers, MA, USA), diluted 1:10,000 in PBST, was added, and incubation at 37 °C was continued for another hour, followed by washing and the addition of the substrate solution, consisting of O-phenylenediamine dihydrochloride (Sigma Aldrich, St. Louis, MO, USA). For the detection of antibody isotypes, specific non-conjugated goat anti-mouse IgG1, IgG2a, IgG2b, and IgG3 or rabbit anti-mouse IgG2c (Sigma Aldrich, St. Louis, MO, USA) diluted at 1:1000 in PBST were used followed by incubation with rabbit anti-goat or goat anti-rabbit-IgG-HRP diluted 1:80,000 in PBST (Bio-Rad Laboratories, Hercules, CA, USA). After the last washing step, the substrate solution was added, and the reaction was stopped by adding 1M HCl. The results were read at 492 nm (OD_492_) using a Multiskan FC spectrophotometer (ThermoScientific, Waltham, MA, USA).

### 2.9. Avidity of Anti-mFhSAP-2 IgG2a and IgG2c Antibodies

The avidity of specific anti-mFhSAP-2 IgG2a and IgG2c antibodies in immunized mice was determined using a urea-based dissociation ELISA, as described by Nazeri et al. (2020) [38]. Two ELISA plates were used in parallel: one for measuring the anti-mFhSAP-2 IgG2a and IgG2c isotypes and the other for measuring their avidity. The coating, blocking, and sample incubation steps were performed as described in the ELISA for total IgG. However, after sample incubation, one plate was washed three times with PBST as usual, while the other plate was washed three times with urea dissociation buffer (8M urea, Thermo Fisher Scientific, Waltham, MA, USA), diluted in PBST, and then washed once with PBST. Following the washes, both plates followed the protocol for detecting specific IgG2a and IgG2c described above. The avidity index (AI) was calculated as the ratio of the OD_492_ of the urea-treated samples to the untreated samples, multiplied by 100. The AI was interpreted as low if <30%, intermediate if between 30% and 50%, and high if >50%.

### 2.10. Splenocyte Proliferation

To assess the ability of mFhSAP-2 to induce T cell proliferation from splenocytes, we used a CyQUANTc Cell Proliferation Assay Kit (Thermo Fisher Scientific, Waltham, MA, USA). This assay uses the green fluorescent CyQUANT^®^ GR dye, which exhibits strong fluorescence when bound to nucleic acids of viable cells, allowing for cell quantification based on fluorescence intensity. Splenocytes from C57BL/6 mice immunized with mFhSAP-2+ISA50, along with negative controls, were thawed, diluted in high-glucose RPMI-1640 supplemented with L-glutamine and HEPES (ATCC, Manassas, VA, USA), and centrifuged at 300× *g*. Then, the cells were washed once, resuspended in RPMI-1640 supplemented with 10% heat-inactivated fetal bovine serum (iFBS), 100 U/mL penicillin, and 100 µg/mL streptomycin, and incubated at 37 °C and 5% CO_2_ for 1 h. After incubation, cell viability was assessed using trypan blue staining. Splenocytes from each mouse were seeded in triplicate at a concentration of 5 × 10^4^ cells/well in a flat-bottom 96-well plate. Cells were then stimulated with 10 µg/mL mFhSAP-2 or left unstimulated for 72 h at 37 °C in 5% CO_2_. Following incubation, the culture medium was carefully removed from each well, and the plate was frozen at −80 °C. A reference curve was also generated on the same plate by serially diluting a cell suspension of 1 × 10^6^ cells/mL, which was also stored at −80 °C. Following the manufacturer’s instructions, on the day of the experiment, the plate was thawed, CyQUANT^®^ GR dye/cell-lysis buffer (diluted 1:80) was added to all wells, and after 2 to 5 min of incubation, absorbance at 480 nm (excitation max) and 520 nm (emission max) was measured. The cell proliferation index (CPI) was calculated as the ratio of the number of cells at 72 h to the number of cells at 0 h, based on the linear standard curve, where fluorescence (*y*) was correlated to cell numbers (*x*) using the formula *y* = A + B*x*.

### 2.11. Flow Cytometry

Flow cytometry was performed to assess the activation status of CD4+ and CD8+ T cells as well as the intracellular secretion of IFNγ and IL-4 in CD4+ T cells. Splenocytes were thawed as described above for the cell proliferation assay. Splenocytes from each mouse were seeded in duplicate at a concentration of 2 × 10^5^ cells/well in a round-bottom 96-well plate. Cells were stimulated with 10 µg/mL mFhSAP-2 for 24 h at 37 °C in 5% CO_2_. Using standard protocols, stimulated cells were stained for fluorescent-activated cell sorter analysis with anti-CD3 BV650 (diluted 1:400), anti-CD4 BV711 (diluted 1:400), anti-CD8a FITC (diluted 1:200), and anti-CD69 BV785 (diluted 1:200) from Biolegend (San Diego, CA, USA). For intracellular staining, cells were first permeabilized and then incubated with an antibody cocktail consisting of anti-IFNγ from BD Biosciences (San Jose, CA, USA) and anti-IL-4 PE/Cy7 from Biolegend (San Diego, CA, USA), both diluted at 1:200 in the cocktail. Data acquisition was performed using a two-laser FACS Celesta flow cytometer (BD Biosciences, San Jose, CA, USA). Data were analyzed using FlowJo software v.10. To evaluate the Th1 response, the ratios of CD4+/CD8+ T cells and IFNγ/IL-4 cytokines among activated T cells were calculated, with higher ratios indicating enhanced T helper cell activity and a stronger Th1 response, respectively.

### 2.12. Statistical Analysis

Data are expressed as mean values and the mean standard error of the mean (SEM). Data were analyzed for normality with the Shapiro–Wilk test, and outliers were identified before statistical analyses. A Student’s *t*-test was used to determine significant differences between the groups, with *p* < 0.05 considered significant. Analysis was performed using GraphPad Prism software (v.10).

## 3. Results

### 3.1. mFhSAP-2 Was Successfully Expressed in E. coli with High Purity and Yield

Transfection of pET30a (+)-6His-GST-mFhSAP-2 into *E. coli* BL21 Star^TM^ DE3 competent bacteria resulted in the production of the recombinant fusion protein with a molecular weight of ~38.7 kDa, as revealed via SDS-PAGE and Western blotting (Appendix A). The optimum expression conditions for the protein were determined and reported in a previous study [29]. After digestion and completion of endotoxin removal, mFhSAP-2 was recovered with ≥85% purity at a concentration of 1.20 mg/mL with endotoxin levels ≤0.19 EU/mg. According to FDA guidelines for LAL testing, endotoxin levels below 0.1 EU/mL are generally acceptable for in vivo studies (https://www.fda.gov/regulatory-information/search-fda-guidance-documents) accessed on 25 March 2025.

### 3.2. mFhSAP-2 Exhibits a Secondary Structure Similar to That Predicted for the Full-Length Protein

Circular dichroism (CD) spectra data were analyzed to estimate the fractional composition of each secondary structure element using the BeStSel^TM^ (2014–2024) web server. The spectrum of mFhSAP-2 (Figure 3) recorded at pH 7.2 and 25 °C revealed a composition of 66.0% α-helix and 34.0% extended or random coil (Table 1). The CD data closely resemble the calculated data for the secondary structure of full-length FhSAP-2 as predicted by the SOPMA server [29].

### 3.3. mFhSAP-2 Induces TNFα Production in RAW264.7 Cells Without Affecting Cell Viability

RAW264.7 cells treated with mFhSAP-2 secreted detectable levels of TNFα at every concentration tested. The TNFα concentration reached values of 432.03 ± 150.16 pg/mL in the culture media when cells were treated with 5 µg/mL mFhSAP-2, and its levels progressively increased to 714.60 ± 25.10, 878.0 ± 127.00, and 1329.7 ± 255.70 pg/mL when cells were treated with mFhSAP-2 concentrations of 10, 15, and 20 µg/mL, respectively (Figure 4A). Negative control (NC) cells treated with PBS did not secrete any detectable levels of TNFα. However, cells treated with LPS secreted TNFα levels of ~1404 ± 125 pg/mL, which were not significantly different from those induced by the highest mFhSAP-2 concentration tested (20 μg/mL) (Figure 4B). Importantly, none of the mFhSAP-2 concentrations tested or LPS treatments affected the viability of RAW264.7 cells, as evidenced by the high OD_450_ values obtained in the XTT assay, which were similar to or higher than those observed for the PBS-treated cells (Figure 4C).

### 3.4. mFhSAP-2 Induces Strong Anti-mFhSAP-2 IgG Response in F. hepatica-Infected and Immunized Animals

As shown in Figure 5, sera from rabbits and mice experimentally infected with *F. hepatica* exhibited a strong total specific IgG response against mFhSAP-2. There were no significant differences (ns) in the average OD_492_ values between the rabbit sera between early (1.288 ± 0.581) and late infection (1.809 ± 0.236) (Figure 5A). The average OD_492_ observed in infected mice was 1.614 ± 0.665, suggesting that both rabbits and mice produced comparable levels of specific IgG antibodies against mFhSAP-2 during active *F. hepatica* infection. Statistically significant differences were found between the mouse group immunized with mFhSAP-2+ISA50 (average OD_492_ = 2.750 ± 0.280) and the infected group (*p* = 0.0003). However, no significant differences were found between the *F. hepatica*-infected and mFhSAP-2-immunized mice. Moreover, the group immunized with mFhSAP-2+ISA50 also showed significantly higher specific IgG levels than the group immunized with mFhSAP-2 alone (*p* = 0.0026, average OD_492_ = 2.086 ± 0.344) (Figure 5B). In contrast, antibody levels in control rabbits (average OD_492_ = 0.007 ± 0.007) and control mice (average OD_492_ = 0.136 ± 0.084) were close to the assay background.

### 3.5. mFhSAP-2 Induces a Predominant Th1 Antibody Response in Immunized Mice and a Predominant Th2 Antibody Response in Infected Animals

The analysis of specific antibody subtypes in mice immunized with either mFhSAP-2 alone or mFhSAP-2+ISA50 revealed the presence of all antibody subtypes (IgG1, IgG2a, IgG2c, and IgG3). Among these, IgG2c was the most abundant, followed by IgG1, IgG3, and IgG2a. As shown in our results, both formulations (mFhSAP-2 and mFhSAP-2+ISA50) induced a similar IgG subtype profile (Figure 6A,B). In the mFhSAP-2+ISA50-immunized group, the average levels of IgG2c were significantly higher than IgG2a (*p* < 0.0001, OD_492_ = 2.238 ± 0.602 and 1.032 ± 0.171, respectively) and were also significantly higher than IgG1 (*p* = 0.0032, average OD_492_ = 1.504 ± 0.095) and IgG3 (*p* < 0.0001, average OD_492_ = 1.031 ± 0.213). In the mFhSAP-2-immunized group, the IgG2a and IgG2c antibody response was not homogeneous. Two of five mice (40%) had lower IgG2a and IgG2c antibody levels than the other three animals. These two mice had average OD_492_ values of 0.461 and 0.756 for IgG2a, whereas the other three mice had average OD_492_ values of 1.511, 1.516, and 1.754. Similarly, the same two animals had notably lower IgG2c levels (OD_492_ = 0.202 and 0.801) than the other three animals, which had average OD_492_ values of 3.08, 3.153, and 3.37 for IgG2c. Thus, when the average OD_492_ for IgG2c and IgG2a values was analyzed for this group, the average level of IgG2c (OD_492_ = 2.122 ± 1.339) was 1.76-fold higher than the average OD_492_ for IgG2a (1.199 ± 0.499). However, no significant differences between IgG2c and IgG2a or between IgG2c levels and those of IgG1 (average OD_492_ = 1.661 ± 0.049) and IgG3 (average OD_492_ = 0.723 ± 0.150) were found (Figure 6B). mFhSAP-2 was prepared in sterile PBS, and all injections were administered following the same standard protocol. None of the animals developed any abscesses at the site of injection or showed any other visible adverse reactions. Therefore, the low antibody response observed in the two mice from the mFhSAP-2-immunized group cannot be associated with differences in the protein’s preparation or administration.

When analyzing the IgG subtypes of *F. hepatica*-infected mice, it was found that IgG1 was the dominant isotype in all samples (average OD_492_ = 1.567 ± 0.557). Levels of IgG1 were 2.85-fold higher than IgG2a (average OD_492_ = 0.549 ± 0.213), 7.87-fold higher than IgG2c (average OD_492_ = 0.199 ± 0.155), and 4.5-fold higher than IgG3 (average OD_492_ = 0.348 ± 0.144), which were statistically significant findings (*p* = 0.034, *p* = 0.004 and *p* = 0.0016, respectively) (Figure 6C). Further analyses of specific IgG subtypes showed that both mFhSAP-2- and mFhSAP-2+ISA50-immunized mice had a lower IgG1/IgG2c ratio (average 0.90 ± 0.066 and 0.64 ± 0.099, respectively), suggesting a mixture of Th1 and Th2 response with a predominance of the Th1 antibody subtype (IgG2c). In contrast, the IgG1/IgG2a ratio for these groups was higher (1.525 ± 0.252 and 1.327 ± 0.493, respectively), indicating a mixed Th1 and Th2 response, with a predominance of the Th2 antibody subtype (IgG1). Additionally, *F. hepatica* infection showed significantly higher IgG1/IgG2a (3.135 ± 0.910) and IgG1/IgG2c (13.79 ± 6.63) ratios than immunization with mFhSAP-2+ISA50 (*p* = 0.0002 and *p* = 0.0053, respectively), which is typical of a polarized Th2 response (Figure 6D).

### 3.6. mFhSAP-2 Induces High-Avidity IgG2a and IgG2c Antibodies with a Predominance of IgG2c over IgG2a in Immunized C57BL/6 Mice

Mice immunized with mFhSAP-2+ISA50 had high-avidity IgG2a and IgG2c antibodies (AI > 50%), while mice infected with *F. hepatica* had intermediate-avidity IgG2a antibodies (AI between 30% and 50%) and low-avidity IgG2c antibodies (AI < 30%) (Figure 7A, Table 2). Moreover, the mean serum antibody titers in both mFhSAP-2- (Figure 7B) and mFhSAP-2+ISA50-immunized groups (Figure 7C) showed lower IgG2a end-point titers (1:1600 and 1:6400, respectively) compared to IgG2c titers (1:3200 and 1:25,600, respectively).

### 3.7. Immunization with mFhSAP-2 Induces Splenocyte Proliferation and Predominance of CD4+ T Cells over CD8+ T Cells with Higher IFNγ than IL-4 Production

Splenocytes from immunized (mFhSAP-2+ISA50) and naïve mice were stimulated ex vivo with mFhSAP-2 or left unstimulated. Splenocytes from immunized mice, which were non-stimulated ex vivo, exhibited an average cell proliferation index (CPI) of 1.809 ± 0.142, which was significantly higher (*p* = 0.0176) than the average CPI of non-stimulated splenocytes from naïve mice. However, splenocytes from immunized mice that were stimulated ex vivo with mFhSAP-2 exhibited a CPI of 1.449 ± 0.138, which was not significantly different from that exhibited by splenocytes from naïve mice stimulated with mFhSAP-2 (1.147 ± 0.091) (Figure 8A). Additionally, the cytokine production and activation status of T helper cells (CD4+) and cytotoxic T cells (CD8+) of mice immunized with mFhSAP-2+ISA50 were determined using flow cytometry. Splenocytes from immunized mice stimulated ex vivo with mFhSAP-2 were labeled with specific antibodies for CD3+, CD4+, CD8+, CD69, IFNγ, and IL-4. The results were compared to those shown by labeled splenocytes from naïve mice stimulated ex vivo with mFhSAP-2. As our results show, immunized mice exhibited significantly higher (*p* = 0.0252) CD4+/CD8+ ratios (16.02 ± 0.734) than naïve mice (9.77 ± 2.940), indicating a predominance of activated CD4+ T cells over CD8+ T cells in immunized mice (Figure 8B). When analyzing the IFNγ/IL-4 ratios in CD4+ T cells, immunized mice had average ratios of 0.950 ± 0.126, which were significantly higher (*p* = 0.0376) than those observed in naïve mice (0.468 ± 0.058) (Figure 8C). In contrast, the levels of IFNγ and IL-4 secreted by CD8+ T cells from immunized mice were similar to those observed in naïve mice.

## 4. Discussion

In the search for potential vaccine candidates against *F. hepatica*, antigens that promote T helper 1 (Th1) immune responses have been proposed as promising targets. This approach appears to contradict the established paradigm that anti-helminth protection depends on the mobilization and activation of various immune cells, including type-2 macrophages and Th2 cells. These cells are known to promote Th2-associated cytokines such as IL-4, IL-5, and IL-9, among others [39]. However, *F. hepatica* infection exhibits strong immunomodulatory behavior that induces a dominant Th2 immune response while actively suppressing the Th1 responses in the host [40,41]. The rationale for developing *F. hepatica* vaccines that promote a Th1 response is based on the studies performed on Indonesian thin-tail (ITT) sheep, which are naturally resistant to *Fasciola gigantica* infection [28,42] and exhibit high levels of specific IgG2a antibodies and Th1-associated cytokines [28]. Furthermore, other investigators assessing the protective capacity of *F. hepatica*-derived antigens, such as secreted proteases, leucine aminopeptidases, Kunitz-type inhibitors, and multimeric compounds, also reported moderate to high levels of protection against *F. hepatica*, associated with elevated levels of specific IgG2a antibodies [18,19,43,44]. Our results from previous studies using the full-length FhSAP-2 antigen are consistent with these findings and demonstrate that FhSAP-2 induces Th1 responses associated with parasite clearance.

FhSAP-2 is a member of the *F. hepatica* saposin-like/NK-lysin protein family [34,37,45]. Although FhSAP-2 has induced substantial protection levels ranging from 60% [34] to more than 80% [37,45], scaling up vaccination trials in larger animal models, such as cattle or sheep, has been largely hindered by challenges in producing the protein in sufficient quantities. This limitation arises from the high hydrophobic character of FhSAP-2 and the low yield typically obtained during its production, even after extraction from inclusion bodies. Its tendency to aggregate and form non-specific interactions with other molecules in solutions makes this protein difficult to purify at high yields. Therefore, the primary objective of this study was to determine whether a modified version of the protein, lacking the first 16 amino acids at the N-terminus, which includes a large string of hydrophobic residues and one of the protein’s two dominant Th1 epitopes, could still induce a Th1-type immune response. Our results demonstrate that omitting this hydrophobic region and adding a synthetic amino acid sequence, such as GST at the N-terminus, significantly improved the protein’s solubility. Upon expressing this modified construct in *E. coli*, the resulting protein, now termed mFhSAP-2, was recovered with a high purity index (>85%) and an acceptable yield.

Given that proper folding of antigenic epitopes is crucial for antibody recognition in vaccine development, we assessed whether mFhSAP-2 could retain antigenic epitopes capable of reacting with antibodies elicited during *F. hepatica* infection. Our results showed that mFhSAP-2 was highly reactive with sera from both rabbits and mice with active *F. hepatica* infection. This suggests that removing the N-terminal region did not compromise antigenic recognition and that proper folding of the protein was preserved. This is consistent with the understanding that antigenic sites in a protein are primarily localized in regions with a low hydrophobicity and high solvent accessibility [46]. Moreover, circular dichroism (CD) analysis revealed that mFhSAP-2 has an ordered secondary structure, consisting of a 66% alpha helix, which closely aligns with the 67.33% alpha helix content predicted for the full-length FhSAP-2 using an independent bioinformatic algorithm (SOPM software) [29]. These observations further suggest that mFhSAP-2 retained its main structural features, comprised of five amphipathic α helical domains, along with six cysteine residues and seven hydrophobic residues at strictly conserved positions [29].

Upon confirming the proper folding of mFhSAP-2, we proceeded to determine whether the protein retained its capacity to induce immune responses both in vitro and in vivo. The initial experiments were conducted using RAW264.7 cells, which, when stimulated with LPS, mimic the behavior of macrophages in vivo by significantly increasing the production of TNFα, a pro-inflammatory cytokine involved in the innate immune response [47,48]. The finding that different concentrations of mFhSAP-2 induced detectable levels of TNFα in RAW264.7 cell culture media, with TNFα production increasing in a dose-dependent manner without affecting cell viability, suggests that mFhSAP-2 is nontoxic and capable of activating macrophages. These in vitro observations, along with the high levels of specific IgG antibodies observed in mice immunized with mFhSAP-2 alone, confirm that mFhSAP-2 is immunogenic even in the absence of an adjuvant. However, this group showed high variability in its antibody response compared to the more homogeneous and robust response observed in mice immunized with mFhSAP-2+ISA50, suggesting that the use of an adjuvant may be required to enhance the immunogenicity and homogenize mFhSAP-2’s response. The adjuvant could enhance the magnitude, breadth, and longevity of a specific immune response and could also influence the quality of the response [49,50,51].

To further explore the type of antibody response induced by mFhSAP-2, we analyzed the relative amounts of IgG subtypes in sera of the immunized animals. Mouse IgG2a and IgG2c are signature Th1-type immunoglobulin subtypes primarily stimulated by Th1-associated cytokines such as IFNγ, and their presence is often used as a marker of a dominant Th1 immune response [52]. In contrast, the IgG1 subtype is considered a Th2-type antibody, meaning its production is primarily associated with Th2 immune responses characterized by the production of Th2-associated cytokines like IL-4 [53]. The IgG2a and IgG2c isotypes are encoded by closely linked genes within the IgG heavy constant region. In mice, these isotypes are inherited as haplotypes Igh-1a and Igh-1b, respectively, and are organized in tandem on the same chromosome as two distinct loci [54]. Since not all mouse strains express both haplotypes at the same time or intensity, it is important to consider the specific mouse strain for an accurate interpretation of the Th1/Th2 response. In our study, we used C57BL/6 mice, which are considered to have a Th1-biased genetic background [55]. According to some authors, C57BL/6 mice could exclusively express IgG2c due to a deletion in the IgG2a gene [54] or express significantly more IgG2c than IgG2a [38]. Our results in C57BL/6 mice immunized with mFhSAP-2 or mFhSAP-2+ISA50 are consistent with studies showing a clear IgG2c-biased subtype response [55]. This was confirmed by the higher IgG2c titers observed in the immunized mice, compared to the lower end-point titers for IgG2a. Therefore, studies measuring IgG2a instead of IgG2c in C57BL/6 mice could draw misleading conclusions about the immune response induced by a recombinant experimental vaccine [56,57].

The observation that two of five animals (40%) in the group immunized with mFhSAP-2 developed very low IgG2a and IgG2c antibody levels compared to the other three animals in the same group suggests that, in the absence of adjuvant, some animals can fail to elicit a robust antibody response. Considering that there were no differences in the preparation of the antigen or pattern of injection, that none of the animals developed abscesses or showed any visible adverse reactions associated with the injections, and that all animals from the group immunized with the mFhSAP-2+ISA50 formulation developed a similar and homogenous antibody response with a predominance of IgG2a and IgG2c antibodies, it is suggested that the use of an adjuvant may be required to enhance the immunogenicity and homogeneity of the mFhSAP-2 antibody response.

The high levels of Th1-associated antibodies induced by mFhSAP-2 formulated with Montanide ISA50 are consistent with previous studies on recombinant vaccines for bovine herpesvirus, malaria, leishmania, and toxoplasmosis, where Montanide ISA50 was used to enhance Th1 immune responses [58,59,60,61]. Importantly, this Th1-associated antibody profile is notably opposed to those typically observed during an active infection with *F. hepatica* [40,41], which may precede high levels of protective immunity to a challenge infection. Unfortunately, due to the lack of a feasible source of *F. hepatica* metacercariae, which is the infective stage of the parasite, we were unable to confirm whether the observed Th1-induced immune response was associated with total or partial parasite clearance, which represents a limitation of the present study. Despite this, the observation that the mFhSAP-2+ISA50 formulation retains its antigenicity and Th1-inducing capabilities suggests that mFhSAP-2 may resemble the full-length protein in its capacity to induce protective immunity. In this regard, the high IgG2 (a/c) antibody titers and levels of IFNγ elicited by mFhSAP-2+ISA50 were also comparable to those elicited by a vaccine based on a Kunitz-type vaccine adjuvanted with CpG-ODN tested in mice, which significantly reduced liver damage and mortality in mice challenged with *F. hepatica* [19]. The antibody and cytokine profiles induced by mFhSAP-2+ISA50 were also comparable to those elicited by an ESP-based vaccine adjuvanted with Naloxon, which achieved 62.5% protection and was associated with high levels of IgG2a and IFNγ [62]. Similarly, comparable antibody and cytokine responses were observed with recombinant SAP-2 and leucine-aminopeptidase homologs from *F. gigantica* (FgSAP-2 and FgLAP), tested in mice either as single antigens emulsified in Freund’s adjuvant [63,64] or as a combined vaccine [65]. These formulations elicited high levels of specific IgG2a antibodies and conferred levels of protection ranging from 60.8% to 78.5% for single-antigen vaccines [63,64] and 80.7 to 81.4% for the combined vaccine [65].

Evaluating antibody avidity is another important factor in vaccine development. Avidity is the strength with which an IgG antibody subtype binds to its specific target epitope. It is established during the affinity maturation process, and disruptions in this process may lead to insufficient protective immunity against infections and diseases [66,67]. In our study, the mouse group immunized with mFhSAP-2+ISA50 exhibited high-avidity IgG2a and IgG2c antibodies, whereas the infected group developed IgG2a and IgG2c antibodies with low to moderate avidity. Since C57BL/6 mice predominantly express IgG2c instead of IgG2a, assessing only IgG2a avidity could lead to misinterpretations of a vaccine candidate’s potential to induce protection against infection. Moreover, previous studies have shown that increased levels of high-avidity IgG2 (a/c) antibodies correlate with lower *F. hepatica* fluke burdens and enhanced immune protection [68]. These antibodies also facilitate pathogen elimination through mechanisms such as antibody-dependent cellular cytotoxicity (ADCC), which plays a critical role in eliminating *F. hepatica* NEJs [14,69]. In our study, both IgG2a and IgG2c isotypes induced by immunization with mFhSAP-2+ISA50 exhibited high avidity, reinforcing their potential role in mediating protective immunity. Additionally, to further determine whether our vaccine induced a Th1 immune response, which is associated with Th1-associated antibodies like IgG2c, we also analyzed T cell proliferation, the activated CD4+/CD8+ T cell populations, and IFNγ/IL-4 production induced by immunization. As expected, a higher cell proliferation index and higher CD4+/CD8+ and IFNγ/IL-4 ratios were observed in mice immunized with mFhSAP-2+ISA50, confirming that this vaccine formulation induces polarized Th1 immune responses.

The promising results obtained with the mFhSAP-2+ISA50 formulation make it possible to plan future vaccination trials in ruminants, which are the main hosts of *F. hepatica.* However, because ruminants are phylogenetically distant from mice, the optimized conditions obtained herein for the mFhSAP-2+ISA50 formulation using C57BL/6 mice may not be directly extrapolated to larger animals. The doses of mFhSAP-2 and the frequency of injections need to be optimized. Moreover, cattle and sheep generally exhibit a Th2-driven immune response that is exacerbated by *F. hepatica* infection [70,71,72], whereas the mouse strain C57BL6 tends to develop Th1 immune responses [55]. Therefore, it is necessary to rule out the possibility that the genetic background of mice influenced the immune responses induced by the mFhSAP-2+ISA50 formulation. Hence, studies prior to the initiation of vaccination trials in ruminants should focus on elucidating whether this formulation could elicit similar Th1-driven immune responses in a mouse strain like BALB/c, which tends to develop Th2 responses [73]. Moreover, if it is demonstrated that our vaccine induces protective immunity in Th2-biased animals by eliciting a response opposite to that observed in natural infection—such as a mixed Th1/Th2 response with a predominance of Th1 response, as observed in our studies with C57BL/6 mice—it will be necessary to assess the longevity, avidity, and affinity of this immune response. Additionally, we will need to determine how these responses may fluctuate depending on the genetic background of the animals, whether repeated doses of the vaccine (monthly or yearly) may be required to prevent natural infection, or whether boosting through natural infection could maintain or even enhance protection. Moreover, if needed, the mFhSAP-2+ISA50 formulation could be optimized by supplementing it with CpG-ODG, which has been shown to potentiate the immunogenicity of small vaccine molecules [19]. Moreover, considering that sheep and cattle models are costly, it would be advisable to gather sufficient efficacy and safety data for the mFhSAP-2+ISA50 vaccine during the lifespan of mice (1 year) before transitioning to these larger models. This includes assessing the durability of IgG2a and IgG2c titers, evaluating the avidity and affinity of these antibodies, identifying any major adverse effects in the lungs, kidneys, or liver, and determining the extent of vaccine-induced immune protection.

## 5. Conclusions and Study Limitations

Our results demonstrated that a modified recombinant version of FhSAP-2 (mFhSAP-2) lacking one of its dominant Th1 epitopes retained its immunogenicity and capability to induce Th1-biased immune responses in C57BL/6 mice. However, our observation that mFhSAP-2 alone may be poorly immunogenic compared to mFhSAP-2+ISA50, based on the high variability observed in the specific antibody response of this group, could have been influenced by the small sample size of this group (*n* = 5), which is a limitation of this study. Further studies with a larger sample size (*n* = 10) could unequivocally confirm these findings. Nonetheless, the observation that mFhSAP-2 emulsified with Montanide ISA50 (*n* = 10) elicited a high and homogeneous antibody response suggests that the antibody responses elicited by mFhSAP-2 can be enhanced through the use of an adjuvant such as Montanide ISA50. While mFhSAP-2+ISA50 induced high-avidity IgG2a and IgG2c antibody subtypes, accurately interpreting a Th1 immune response in the C57BL/6 mouse strain is more reliable when measuring IgG2c rather than IgG2a, alongside IFNγ production. The antibody profile elicited by mFhSAP-2+ISA50 was the opposite of that observed in *F. hepatica*-infected mice, which displayed significantly higher IgG1 levels than IgG2c levels, characteristic of a Th2-polarized immune response. However, the inability to perform a challenge infection experiment in the immunized mice was another limitation of this study. Moreover, since C57BL6 is a mouse strain that tends to develop Th1-biased responses, it is necessary to ascertain whether the mFhSAP-2+ISA50 formulation could also induce a Th1 immune response in BALB/c mice, which tend to develop Th2 responses. This would have allowed us to determine whether the observed Th1-biased immune response induced by mFhSAP-2+ISA50 vaccination could be independent of the genetic background of animals and correlates with parasite clearance, decreased egg output, and/or a reduction in liver pathological damage. Furthermore, the optimized genetic construct encoding mFhSAP-2, along with the refined expression and purification protocol developed in this study, enabled the production of mFhSAP-2 with high purity and protein yield. This advancement facilitates the expansion of vaccination trials to larger animal models of veterinary importance.

## Figures and Tables

**Figure 1 vaccines-13-00545-f001:**
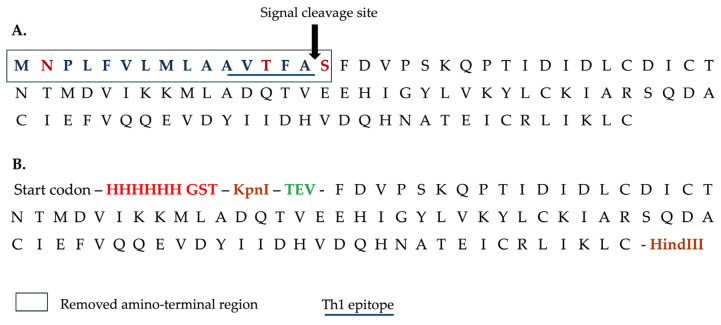
Strategy for cloning a modified version of recombinant FhSAP-2. (**A**) Sequence of 101 amino acids of FhSAP-2 (11.5 kDa) as reported in GenBank (Accession No. AF286903). The box over the first 16 amino acids represents the removed amino terminal region in mFhSAP-2, containing hydrophobic (blue) and hydrophilic (red) residues, along with a predicted signal peptide cleavage site between Ala^15^ and Ser^16^ (arrow). The amino acid sequence ^11^AVTFA^15^ (underlined) represents one of the two dominant Th1 epitopes of FhSAP-2. (**B**) The mFhSAP-2 fusion protein sequence lacks the initial 16 amino acids and the dominant Th1 epitope. A 6His tag and a GST tag (red) were added at the amino terminus to facilitate detection and purification, respectively, and Kpnl and Hind-III restriction sites (brown) were added to mFhSAP-2 for cloning into the pET30a (+) vector. A TEV protease cleavage site (green) was also added at the amino terminus to allow the removal of the His-GST tag after purification.

**Figure 2 vaccines-13-00545-f002:**
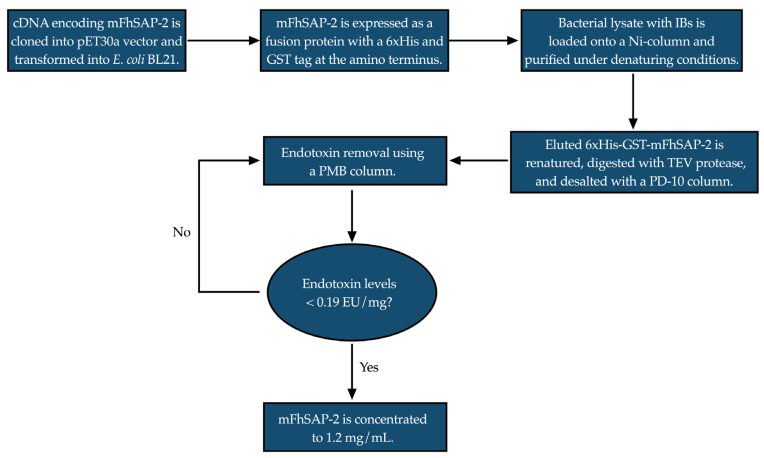
Flowchart showing a simplified sketch of the process for the expression and purification of mFhSAP-2. IBs: Inclusion bodies. Arrows in the figure indicate the order in which each step in the process is accomplished.

**Figure 3 vaccines-13-00545-f003:**
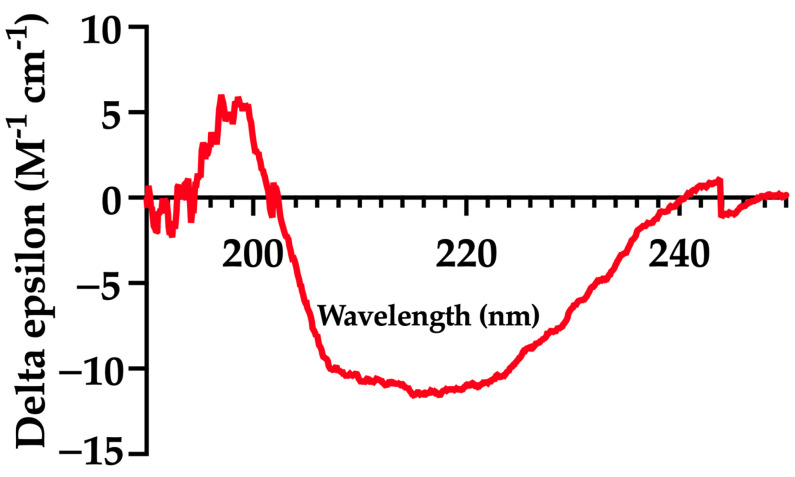
Circular dichroism (CD) spectra of purified mFhSAP-2. The red line represents the CD spectrum, which was recorded using a Jasco J-1500 spectropolarimeter at a concentration of 200 μg/mL at 25 °C, over a wavelength range of 190 and 250 nm. CD raw data were analyzed using the BeStSel^TM^ web server.

**Figure 4 vaccines-13-00545-f004:**
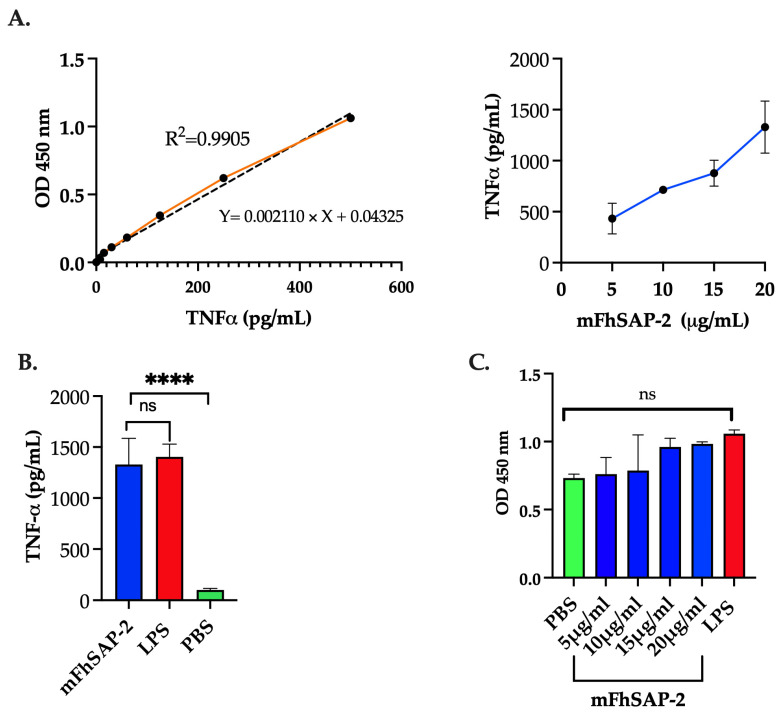
TNFα secretion and cell viability in RAW 264.7 cells treated with mFhSAP-2. RAW264.7 cells are macrophage-like cells of murine origin. Cells were seeded in an RPMI medium at a concentration of 5 × 10^5^ cells/mL and stimulated with increasing concentrations of mFhSAP-2 (5 to 20 μg/mL). Cells stimulated with LPS (10 μg/mL) were used as an activation control, and cells treated with PBS were used as a negative control. (**A**) Orange line represent the OD 450nm values for each TNFα concentration of the standard curve. Dashed line shows the lineal correlation between known TNFα concentrations (*x*-axis) and OD_450_ values (*y*-axis). (**B**) Dose–response curve (showed in blue color) showing TNFα secretion in response to mFhSAP-2 in RAW 264.7 cells. TNFα secretion in RAW 264.7 cells treated with mFhSAP-2 (20 μg/mL), LPS (10 μg/mL), and PBS (negative control). (**C**) Cell viability assay in RAW 264.7 cells treated with PBS, mFhSAP-2, or LPS (10 μg/mL). OD values equal to or above PBS (negative control) indicate high cell viability and metabolic activity. Data are presented as mean ± SEM. Statistical significance was determined using unpaired *t*-tests: **** *p* < 0.0001 and ns: not significant.

**Figure 5 vaccines-13-00545-f005:**
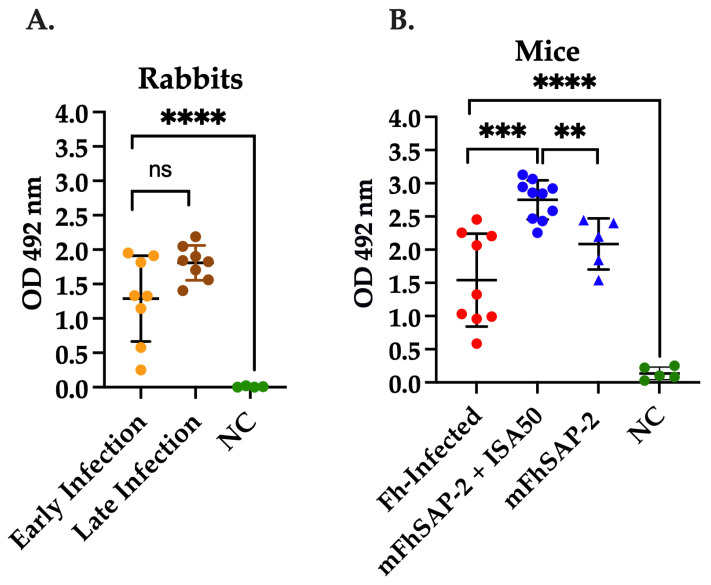
Specific IgG antibodies against mFhSAP-2 in NZW rabbits and C57BL/6 mice. The ability of mFhSAP-2 to react with sera from infected and immunized animals was measured using ELISA. (**A**) Levels of specific anti-mFhSAP-2 IgG antibodies elicited in New Zealand White (NZW) rabbits during early and late stages of *Fasciola hepatica* challenge infection. Rabbit sera collected prior to infection were used as the negative control (NC). (**B**) Levels of specific anti-mFhSAP-2 IgG antibodies elicited in C57BL/6 mice infected with *F. hepatica* or immunized either with mFhSAP-2+ISA50 or with m-FhSAP-2 alone. Sera from naïve C57BL/6 mice were used as negative controls (NC). Data are presented as mean ± SEM. Statistical significance was determined using unpaired *t*-tests: **** *p* < 0.0001, *** *p* = 0.0003, ** *p* = 0.0026, and ns: not significant.

**Figure 6 vaccines-13-00545-f006:**
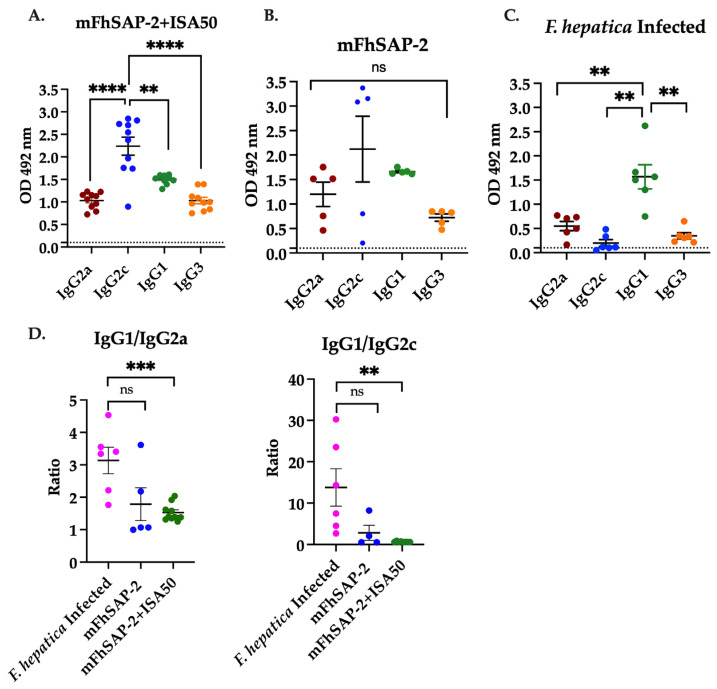
IgG subtype profile induced by mFhSAP-2 in C57BL/6 mice compared to *F. hepatica* infection. ELISA was used to assess the IgG profile against mFhSAP-2 in C57BL/6 mice immunized with mFhSAP-2 or mFhSAP-2+ISA50 or infected with *F. hepatica*. (**A**) mFhSAP-2+ISA50 induced significantly higher IgG2c levels than IgG2a (**** *p* < 0.0001), IgG1 (** *p* = 0.0032), or IgG3 (**** *p* < 0.0001). (**B**) mFhSAP-2 also induced higher IgG2c levels than other subtypes, but the differences were not statistically significant (ns). (**C**) *F*. *hepatica* infection induced significantly higher IgG1 levels than IgG2a (** *p* = 0.0034), IgG2c (** *p* = 0.0040), or IgG3 (** *p* = 0.0016). Dashed lines in (**A**–**C**) indicate the cut-off values (OD > 0.1) for all IgG isotypes established by the average OD_492_
± 3 SD of the negative control animals. (**D**) *F. hepatica*-infected mice had significantly higher IgG1/IgG2a (*** *p* = 0.0002) and IgG1/IgG2c (** *p* = 0.0053) ratios than those immunized with mFhSAP-2+ISA50, whereas differences with those immunized with mFhSAP-2 alone were not significant (ns). Data are presented as mean ± SEM. Statistical significance was determined using unpaired *t*-tests. Dotter line in the figures (**A**–**C**) represent the positive cut-off value for each antibody isotype.

**Figure 7 vaccines-13-00545-f007:**
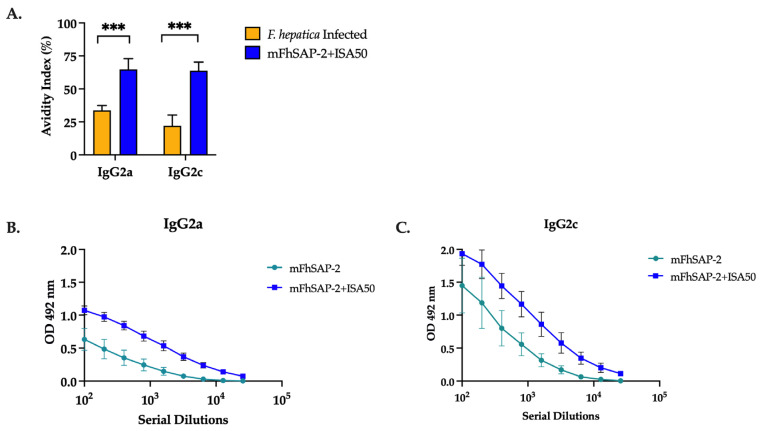
IgG2a and IgG2c avidity index and titration curves. The avidity and titers of anti-mFhSAP-2 IgG2a and IgG2c antibodies were measured using ELISA. (**A**) Avidity index of IgG2a and IgG2c antibodies in C57BL/6 mice either infected with *F. hepatica* or immunized with mFhSAP-2+ISA50. Avidity was interpreted as low if AI < 30%, intermediate if between 30% and 50%, and high if >50%. Immunization with mFhSAP-2+ISA50 induced high-avidity IgG2a and IgG2c antibodies that were significantly higher than those induced by *F. hepatica* infection (*** *p* = 0.001), which elicited moderate-to-low-avidity IgG2a and IgG2c antibodies, respectively. (**B**,**C**) The mean serum antibody titers showed that both mFhSAP-2- and mFhSAP-2+ISA50-immunized mice had lower IgG2a end-point titers (1:1600 and 1:6400, respectively) compared to IgG2c (1:3200 and 1:25,600, respectively). Data are presented as mean ± SEM.

**Figure 8 vaccines-13-00545-f008:**
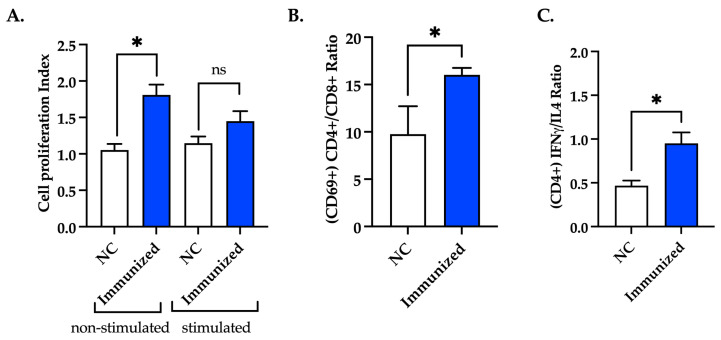
Activation of CD4+ T cells in response to mFhSAP-2. (**A**) Splenocytes from C57BL/6 mice immunized with mFhSAP-2+ISA50 or naïve (negative control: NC) were cultured in vitro, either non-stimulated or stimulated with mFhSAP-2 for 72 h. The cell proliferation index (CPI) was calculated as the ratio of the number of cells at 72 h to the number of cells at 0 h. Non-stimulated cells from immunized mice showed a significantly higher proliferation index than those from negative controls (* *p* = 0.0176). (**B**,**C**) Splenocytes from C57BL/6 mice immunized with mFhSAP-2+ISA50 or naïve (negative control: NC) were cultured in vitro, stimulated with mFhSAP-2 for 24 h, and analyzed using flow cytometry. Mice immunized with mFhSAP-2+ISA50 had a significantly higher number of activated (CD69+) CD4+ T cells (* *p* = 0.0252) than activated CD8+ T cells compared to naive animals. CD4+ T cells from mFhSAP-2+ISA50-immunized mice showed significantly higher IFNγ/IL-4 ratios than NC animals (* *p* = 0.0376). Data are presented as mean ± SEM. Statistical significance was determined using unpaired *t*-tests.

**Table 1 vaccines-13-00545-t001:** Secondary structure analysis of mFhSAP-2.

Method	α-Helix (%)	β-Sheet (%)	Extended or Random Coil (%)
Circular dichroism (25 °C)	66.00	-	34.00
SOPMA Server *	67.33	3.96	18.71

* Prediction made for the full-length FhSAP-2 protein in Espino, A.M., Hillyer, G.V. 2003, J. Parasitol. 89 (1): 545–552 [29].

**Table 2 vaccines-13-00545-t002:** Avidity analysis of IgG2a and IgG2c antibodies to mFhSAP-2 + ISA50.

	Avidity Index (% ± SEM)	
Group	IgG2a	IgG2c
1-*F. hepatica*-infected	42.55% ± 1.46	21.45% ± 15.84
2-mFhSAP-2+ISA50-immunized	72.73% ± 23.01	66.16 ± 19.18

## Data Availability

All data are presented in this manuscript and provided as Appendix A.

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
