# Peer review of "A Modified Variant of Fasciola hepatica FhSAP-2 (mFhSAP-2) as a Recombinant Vaccine Candidate Induces High-Avidity IgG2c Antibodies and Enhances T Cell Activation in C57BL/6 Mice"

_vaccines, 2025, doi:10.3390/vaccines13050545_

Round 1
Reviewer 1 Report (Previous Reviewer 1)
Comments and Suggestions for Authors
My comments have been addresses
Author Response
Dear Reviewer,
Authors greatly appreciate your recommendations and we are happy our responses have addressed all your concerns and questions.
Reviewer 2 Report (Previous Reviewer 2)
Comments and Suggestions for Authors
The present study (ID:vaccines-3623219) titled "A modified variant of Fasciola hepatica FhSAP-2 (mFhSAP-2) as a recombinant vaccine candidate induces high-avidity IgG2c antibodies and enhances T-cell activation in C57BL/6 mice" shows the redesigning of a vaccine protein (FhSAP-2) from the liver fluke parasite Fasciola hepatica to make it easier to purify and tested whether it still provokes strong immune responses in mice. The study was conducted meticulously, but it still needs a lot improvement.
Abstract. Abstract is very important section and should be very compact. This section is over crowded with details, it clutters the main message.
Introduction. This section shows some repetition. Similar information about vaccine candidates and immune evasion is repeated, making the section unnecessarily long.
Materials and Methods. This section has an excessive technical detail (e.g., exact compositions of buffers and enzyme treatments) that interrupts narrative flow. The complexity of the methods would be easier to follow with visual summaries. Some information should be moved to supplementary files.
Results. Inconsistent immune responses in a few mice are mentioned but were not explored or explained.
Discussion. Potential limitations (e.g., species differences, experimental scale) should be adequately addressed. Future directions should be more clearly mentioned in details.
English language. The English is generally understandable but not polished. Scientific terms are used correctly; there’s no major scientific misunderstanding caused by language. there are some grammatical errors and typos. For example:
- "affect" is used instead of "affects" ("Fasciolosis also affect over 2.4 million people...")
- "imunized" instead of "immunized"
- Missing words ("high costs associated to thid treatment" → should be "associated with this treatment")
Author Response
REVIEWER # 2
The present study (ID:vaccines-3623219) titled "A modified variant of Fasciola hepatica FhSAP-2 (mFhSAP-2) as a recombinant vaccine candidate induces high-avidity IgG2c antibodies and enhances T-cell activation in C57BL/6 mice" shows the redesigning of a vaccine protein (FhSAP-2) from the liver fluke parasite Fasciola hepatica to make it easier to purify and tested whether it still provokes strong immune responses in mice. The study was conducted meticulously, but it still needs a lot improvement.
Abstract. Abstract is very important section and should be very compact. This section is over crowded with details, it clutters the main message.
Reply. Authors appreciate your comments, which contribute to improve our manuscript. The abstract was compacted and excessive details were removed.
Introduction. This section shows some repetition. Similar information about vaccine candidates and immune evasion is repeated, making the section unnecessarily long.
Reply. The repetitions related to the vaccine candidates and immune evasion were removed
Materials and Methods. This section has an excessive technical detail (e.g., exact compositions of buffers and enzyme treatments) that interrupts narrative flow. The complexity of the methods would be easier to follow with visual summaries. Some information should be moved to supplementary files.
Reply. Because most of technical details provided were from standard procedures or from manufacturer’s instructions these were removed.
Results. Inconsistent immune responses in a few mice are mentioned but were not explored or explained.
Reply. The inconsistence about the immune responses in a few mice was explained in the discussion section. A paragraph that read as follow was included:
“The observation that 2 of 5 animals (40%) of the group immunized with mFhSAP-2 developed very low IgG2a and IgG2c antibody levels compared to the other 3 animals of the same group suggest that in the absence of adjuvant some animals can fail in eliciting robust antibody response. Considering that there was no differences in the preparation of the antigen, or pattern of injection, none of the animals developed abscesses, or showed any visible adverse reactions associated with the injections and that all animals from the group immunized with the mFhSAP-2+ISA50 formulation developed a similar and homogenous antibody response with predominance of IgG2a and IgG2c antibodies, suggests that the use of an adjuvant may be required to enhance the immunogenicity and homogeneity of the mFhSAP-2 antibody response”.
Discussion. Potential limitations (e.g., species differences, experimental scale) should be adequately addressed. Future directions should be more clearly mentioned in details.
Reply. In the previous version we had mentioned that due to the large phylogenetic distance between mice and ruminants it was not possible extrapolate to ruminants the optimized conditions previously stablished for mice. It is necessary to optimize in larger animals the dose and frequency of injections. In the revised version we also added to the discussion the need to rule out the influence of genetic background of animals. We mentioned that ruminants, specifically cattle and sheep, generally exhibit a Th2-driven immune response that is exacerbated by F. hepatica infection, whereas the mouse strain C57BL6 tend to develop Th1-immune responses. Therefore, prior to initiate vaccination trials in ruminants more studies need to be performed to elucidate whether this our formulation could elicits similar Th1-driven immune responses in a mouse strain like BALB/c, which tend to develop Th2-responses. Additionally, we will need to determine how these responses may fluctuate depending on the genetic background of the animals, whether repeated doses of the vaccine (monthly or yearly) may be required to prevent natural infection or if boosting through natural infection couldy maintain or even enhance protection. The need to ruleout the influence of genetic background of animals in the immune responses induced by mFhSAP-2+ISA50 was also mentioned in the conclusion and Limitations section.
English language. The English is generally understandable but not polished. Scientific terms are used correctly; there’s no major scientific misunderstanding caused by language. there are some grammatical errors and typos. For example:
- "affect" is used instead of "affects" ("Fasciolosis also affect over 2.4 million people...")
- "imunized" instead of "immunized"
- Missing words ("high costs associated to thid treatment" → should be "associated with this treatment")
Reply. These mistakes were corrected. Moreover, the manuscript was subjected to revision by the MDPI editing services.
Reviewer 3 Report (Previous Reviewer 3)
Comments and Suggestions for Authors
The authors have improved the manuscript significantly and addressed all my questions and comments, so I have no more significant comments or questions. However, there are several typos that appear throughout the text of the article
Minor typos: line 93, lines 224, 225, line 454
Author Response
Dear Reviewer,
Dear Reviwer,
Authors greatly appreciate your recommendations and we are happy our responses have addressed all your concerns and questions. Our manuscript will be revised by MDPI English editing system to correct all typos and grammatical error still on it.
Round 2
Reviewer 2 Report (Previous Reviewer 2)
Comments and Suggestions for Authors
The authors have satisfactorily responded to the comments in the revised manuscript.
This manuscript is a resubmission of an earlier submission. The following is a list of the peer review reports and author responses from that submission.
Round 1
Reviewer 1 Report
Comments and Suggestions for Authors
Main points.
- This paper presents data from only three groups of mice. The group sizes and 5, 5 and 10. No protection data is presented. The main conclusions are extrapolated from immune analyses of the sera generated. Many of these data are heteroscedastic and I question the integrity of the statistical analysis conducted on this basis. Whilst I am aware that a protection study is probably being planned, the complete lack of any data demonstrating biological activity of the sera against the target pathogen severely compromises the paper. Could a pathogen killing assay be conducted in vitro? It is my opinion that without data demonstrating pathogen killing, the paper is inherently flawed.
- The use of Freunds complete adjuvant is banned in several countries.
- Under ARRIVE guidance, papers that do not follow the guidance should be rejected.
- This review took longer than anticipated because there were so many corrections required in the draft text (46 observation below). In some instances, words were missing and sentences did not make sense. This gives the reviewer the impression that the paper was compiled in haste and induces a high level of vigilance of the data contained within it.
Other points
- Line 15 states that the proposed vaccine antigen was administered as an emulsion and inclusion body but the material and methods does not distinguish which data sets were generated using emulsion and which with inclusion bodies. Please clarify.
- Line 25 and many other instances; In vitro is not in italics
- Lines 26 and 27 appear to missing the word IgG2c as the sentence does not make sense without two additions.
- Line 30. Titre is expressed as a dilution factor but the text assumes that titre is being expressed as a reciprocal of the dilution factor (eg 1:6,400 is higher than 1:1,600). Either explain that titre is the reciprocal of the dilution or change the accompanying text.
- Line 47 delete second comma
- Line 51 states “resistance” is induced but I think this should be an “immune response that results in pathogen clearance”.
- Lines 52 to 54. This sentence needs to be rewritten as it is not clear in meaning as written.
- Line 54-56. Add reference and clarify if this was an immune response that resulted in complete pathogen clearance or not.
- Line 63 implies only partial immune response has occurred but this should be changed to “an immune response resulting in partial clearance of infection.” It may be a very strong humoral immune response but insufficient cellular response for example
- Line 65 states that CFA is a suitable adjuvant for research but this adjuvant is banned in several countries. From a UK perspective the CFA generated data should not be published. Please comply with the ARRIVE guidance.
- Line 65 change the word “their” to “the”
- Line 86 “in vitro” is only partially in italics
- Line 87 “in vivo” not in italics
- Line 101 add “,” after “refication”
- Line 121 change “bellow” to “below”
- Lines 150 to 155. This section contains inadequate detail. Please specific which licence the live challenge was conducted with. Please clarify if there was any health monitoring of animals during and after immunisation. The data suggests that immune responses were not homogenous in the groups with some seemingly non-responders and others not. As the three Fruend’s adjuvant doses were all administered in the scruff, what adverse events took place? Did the strong responders have sterile abscesses? Were adverse events checked for? Why were the three doses not given at different sites. This lack of detail threatens the integrity of the project.
- Line 160. Was the FBS heat inactivated or not? Where were the FBS and two antibiotics sourced from?
- Line 171. I assume this incubation was the overnight incubation in step 2.4? If so, clarify in the text.
- Line 181. As stated above, what licence number was used to authorize this study and what adverse events were observed?
- Line 183 In what matrix were the parasites administered orally?
- Lines 186-190. It should be clearly stated whether or not rabbits and mice completely cleared or partially cleared infection. This is currently not clear. Were there any adverse effects seen? Did the animals loose weight or develop signs of infection? Please clarify in detail. Did this differ between mouse strains? This is critical to the hypothesis of the paper.
- Line 194. Did sterile abscesses appear? Why was the dose administered in the same location? Were the animals restrained by the scruff during handling? Were there any adverse events associated with immunisation and if so, was it in all mice?
- Line 201. Was the sera stored for more than three hours? At what temperature were they stored? Were the sera left to clot at 37 degree C or put into SST tubes? Please add all of these details.
- Line 222. Was the 30% peroxide solution a volume to volume dilution in water of a stock solution? If so, please state molarity of stock solution. Express concentration as molarity, not % v/v.
- Line 224. How long were the plates left to incubate for?
- Line 225. Express concentration as molarity
- Line 246 “et al” not in italics
- Line 356 to 358. This statement is not justified. These determinations were single point, not serial dilutions (which would have provided titers) so the outcome does not justify this conclusion.
- Figure 4B. Did these mice fully clear the pathogen? Were the source sera taken early or late?
- Line 395 does not currently mention IgG2A. Please add missing information.
- Section 3.5 states that the outputs from these analyses exhibited large variances. I am concerned that the statistical analysis of such heteroscedastic data is not sound.
- Line 403. The variance of this data is very high indeed. Were there differences at the injection sites seen in these animals? Could abscesses account for the variance?
- Figure 5 A, B and C are currently bar charts. Section 3.5 clearly states that these data are heteroscedastic. Please change these graphs to scatterplot to reveal this variance in data. Are the multiple cross-tests sound? Do they assume homoscedasity?
- Figures 5 A, B and C. What are the cut-offs for each graph? They are not likely to be zero.
- Line 454. Delete extra space in title
- Line 457 states “non-stimulated, immunized mice” then on line 458 “nonstimulated”. This sentence does not make sense to me.
- Line 460. What is the meaning of stimulated? Do mean immunized? If so, clarify. If not, also clarify
- Line 465 Italics missing
- Line 466 – clarify meaning
- Line 468 Italics missing
- Line 523, delete comma
- Line 524, add “s” to suggest.
- Line 534 Correction required second word
- Line 542 Italics missing
- Line 544 Repetition of words – sentence does not make sense
- References; Many references have italics missing for species names
Many changes needed
Author Response
Response to the Reviewer # 1
General Comments
- This paper presents data from only three groups of mice. The group sizes and 5, 5 and 10. No protection data is presented. The main conclusions are extrapolated from immune analyses of the sera generated. Many of these data are heteroscedastic and I question the integrity of the statistical analysis conducted on this basis. Whilst I am aware that a protection study is probably being planned, the complete lack of any data demonstrating biological activity of the sera against the target pathogen severely compromises the paper. Could a pathogen killing assay be conducted in vitro? It is my opinion that without data demonstrating pathogen killing, the paper is inherently flawed.
Response. The authors greatly appreciate your effort in revising our manuscript and regret that parts of the manuscript were difficult to understand. FhSAP-2 is a vaccine candidate that has previously been shown to induce partial protection in experimental animals challenged with F. hepatica infection (Espino & Hillyer 2004, Espino & Rivera, 2010 and Rivera & Espino, 2016). The focus of the present study was to determine whether a modified version of FhSAP-2 could be as immunogenic and capable of eliciting Th1-biased antibody and cellular responses as the full-length protein. However, determining whether such a Th1-biased immune response is capable of inducing protection against a challenge infection was beyond the scope of the present study. We respectfully disagree that without protection data, this work is inherently flawed as there are numerous articles that characterize the immunological and biochemically properties of antigens, whose value as vaccine candidates or immunodiagnostic antigens is further assessed in subsequent studies. The lack of an F. hepatica immunized-challenged group or an in vitro ADCC experiment in this study is due to our current limited access to a feasible source of metacercariae (mc). The only supplier of mc for scientists in the USA and Europe was Baldwin Aquatic, Inc. (Monmouth, Oregon), which unfortunately ceased operations more than a decade ago. While there are some local producers of mc in South America (i.e. Peru, Argentina), they do so for their own use and not for commercial supply. To continue expanding our vaccinations studies, we are in the process of establishing partnerships with investigators in South America to test the efficacy of our mFhSAP-2 formulation in sheep or cattle. These collaborative studies are still in the preliminary phase, as we are in the process of securing the necessary funding to carry them out. In the revised version of our manuscript, we have highlighted this issue as a limitation of the study.
On the other hand, we respectfully disagree with the suggestion that our conclusions have been based on data that follow a heteroscedastic pattern. In contrast, most of our data followed a homoscedastic pattern. The only data in which high variability resulted in no statistical differences was the group immunized with mFhSAP-2 without adjuvant (Fig. 6B). Two of 5 animals (40%) of this group developed very low IgG2a and IgG2c antibody levels compared to the other 3 animals of the same group. There were no differences in the preparation of the antigen, or pattern of injection, and none of the animals developed abscesses, or showed any visible adverse reactions associated with the injections. Each individual responded uniquely to the same antigen, but often when antigens are small they could fail to elicit robust antibody responses. However, all animals from the group immunized with the mFhSAP-2+ISA50 formulation developed a similar and homogenous antibody response with predominance of IgG2a and IgG2c antibodies, suggesting that the use of an adjuvant may be required to enhance the immunogenicity and homogeneity of the mFhSAP-2 antibody response.
In section 3.5 of the revised version, the following statement was added: “mFhSAP-2 was prepared in sterile PBS, and all injections were administered following the same standard protocol. None of the animals developed any abscesses at the site of injection, or showed any visible adverse reactions. Therefore, the low antibody response observed in two mice from the mFhSAP-2 immunized group cannot be attributed to differences in the protein’s preparation or administration.”.
In the discussion, the following statement was added:
“However, this group showed high variability in its antibody response compared with the more homogeneous and robust response observed in mice immunized with mFhSAP-2+ISA50, suggesting that the use of an adjuvant may be required to enhance both the immunogenicity and homogeneity of mFhSAP-2. The adjuvant could enhance the magnitude, breadth, and longevity of the specific immune response, as well as influence the quality of the response [49-51]. Moreover, in section 5 (conclusions) limitations regarding the small size of this group were mentioned as limitations of the study.
- The use of Freunds complete adjuvant is banned in several countries.
Response. Yes, FCA is banned for human use in several countries, including USA. However, in USA, FCA can be used under specific regulations for research purposes, provided that the study has been rigorously reviewed and approved by the Institutional Animal Care and Use Committee (IACUC) (https://oacu.oir.nih.gov/system/files/media/file/2022-04/b8_adjuvants.pdf), as was the case in our studies with the full-length FhSAP-2 published in 2010. However, it should be highlighted that in the present study, we did not use FCA or inclusion bodies. In fact, the main premise of the present study was to assess the immunogenicity of mFhSAP-2 formulated with an adjuvant that can be used without restrictions in both human and veterinary applications, such as Montanide ISA50. We confirm that in our present study, as well as those previously published, were carried out in compliance with IACUC regulations, which are analogous to the ARRIVE guidelines.
- Under ARRIVE guidance, papers that do not follow the guidance should be rejected.
Response. We agree with this statement, however it does not apply to our study. Both our past and present studies strictly followed IACUC guidelines, and the present study did not include the use of FCA.
- This review took longer than anticipated because there were so many corrections required in the draft text (46 observation below). In some instances, words were missing and sentences did not make sense. This gives the reviewer the impression that the paper was compiled in haste and induces a high level of vigilance of the data contained within it.
Response. We regret that parts of the manuscript were difficult to understand. We have carefully revised this new version, correcting issues related to italics, commas, and missing words throughout the text, and we greatly appreciate these observations. However, some of the comments and questions raised were surprising to us, as we believe may be due to misinterpretations of the manuscript content. Below, we provide our responses to all observations.
Other Points
- Line 15 states that the proposed vaccine antigen was administered as an emulsion and inclusion body but the material and methods does not distinguish which data sets were generated using emulsion and which with inclusion bodies. Please clarify.
Response. In Line 15 we were providing preliminary data regarding the full-length FhSAP-2, which is pertinent for this study. In those preliminary studies, FhSAP-2 was administered as an emulsion with FCA/IFA or as inclusion bodies. However, the present study is not about the full-length protein but about a modified version of this molecule (mFhSAP-2), administered either alone or emulsified in Montanide ISA50. Therefore, it was not necessary to include details about the use of FCA or inclusion bodies in the Materials and Methods section, as they were not used in the present study.
- Line 25 and many other instances; In vitro is not in italics
Response. In vitro is now in italic throughout the manuscript.
- Lines 26 and 27 appear to be missing the word IgG2c as the sentence does not make sense without two additions.
Response. The word IgG2c was added twice to ensure the sentence reads clearly and makes sense.
- Line 30. Titre is expressed as a dilution factor but the text assumes that titre is being expressed as a reciprocal of the dilution factor (eg 1:6,400 is higher than 1:1,600). Either explain that titre is the reciprocal of the dilution or change the accompanying text.
Response. The text refers to the dilution factor, not the reciprocal of the dilution. Indeed, 1:6,400 is higher than 1:1,600. According to the text the 1: 6,400 dilution corresponds to the IgG2a titer elicited by the mFhSAP-2 +ISA50 formulation, while the 1:1,600 corresponds to the IgG2a titer elicited by mFhSAP-2 without adjuvant. This explains why the 1:6,400 dilution is higher than 1:1,600, suggesting that the adjuvanted formulation induced higher IgG2a titers than the formulation without adjuvant, as was also observed for IgG2c. The text read as follow: “Mice immunized with mFhSAP-2+ISA50 developed higher IgG2a and IgG2c titers (1:6,400 and 1:25,600, respectively) than mice immunized with mFhSAP-2 alone (1:1,600 and 1:3,200, respectively).
NOTE: The questions 5 to 13 correspond to the Introduction. Based on the recommendations of another reviewer, we have rewritten the introduction. Therefore, your corrections to the Introduction do not fit exactly to the new version. However, we responded all your queries as follow:
- Line 47 delete second comma
Response. No second comma was found on line 47.
- Line 51 states “resistance” is induced but I think this should be an “immune response that results in pathogen clearance”.
Response. We believe that “resistance” is the correct term in this context, as it is consistent with the term used by the authors of the cited studies (No. 30 & 41). These authors demonstrated that Indonesian thin-tail sheep exhibit natural resistance to Fasciola infection, which is characterized by high levels of IgG2a and IFNg. Therefore, we have used the term "resistance" to align with these previous studies.
- Lines 52 to 54. This sentence needs to be rewritten as it is not clear in meaning as written.
Response. The introduction was rewritten as mentioned above.
- Line 54-56. Add reference and clarify if this was an immune response that resulted in complete pathogen clearance or not.
Response. The introduction was rewritten as mentioned above.
- Line 63 implies only partial immune response has occurred but this should be changed to “an immune response resulting in partial clearance of infection.” It may be a very strong humoral immune response but insufficient cellular response for example.
Response. The introduction was re-written as mentioned above.
- Line 65 states that CFA is a suitable adjuvant for research but this adjuvant is banned in several countries. From a UK perspective the CFA generated data should not be published. Please comply with the ARRIVE guidance.
Response. We understand and respect the UK perspective regarding the use of CFA. However, our previous studies, published in 2010 and 2016, were conducted in compliance with IACUC guidelines in the USA. In the present study, we did not use CFA for immunizing the animals. Additionally, numerous studies published on PubMed have used CFA and/or IFA as adjuvants in various animal models for several medical conditions, including an experimental vaccine against SARS-CoV-2 (please see: doi: 10.1016/j.vaccine.2005.10.046, doi: 10.1186/s40425-019-0625-x, doi: 10.1016/j.micpath.2022.105687).
- Line 65 change the word “their” to “the”
Response. The change was done.
- Line 86 “in vitro” is only partially in italics
- Line 87 “in vivo” not in italics
Response. In vivo and In vitro are now in italic throughout the text,
- Line 101 add “,” after “refication
Response. We were not able to find this mistake.
- Line 121 change “bellow” to “below”
Response. Change was done.
- Lines 150 to 155. This section contains inadequate detail. Please specific which licence the live challenge was conducted with. Please clarify if there was any health monitoring of animals during and after immunization. The data suggests that immune responses were not homogenous in the groups with some seemingly non-responders and others not. As the three Fruend’s adjuvant doses were all administered in the scruff, what adverse events took place? Did the strong responders have sterile abscesses? Were adverse events checked for? Why were the three doses not given at different sites. This lack of detail threatens the integrity of the project.
Response. We are confused by this comment, as we don’t fully understand what this refers to. The lines 150 to 155 belong to section 2.2 (Circular Dichroism) of Materials and Methods, but if you are referring to section 2.3 (Animals), we believe this comment may not be applicable.
In our study, we don’t need any specific license for a live infection challenge, as no live challenge was done in this study. Furthermore, since CFA was not used, we were not concerned with the development of sterile or non-sterile abscesses, which can sometimes occur when CFA is used.
The description of how immunization was done was provided in section 2.8 (Immunization of C57BL/6 mice with mFhSAP-2). Every single subcutaneous (sc) immunization was applied on different sites of the dorsal surface of animals, with either mFhSAP-2 alone (diluted in sterile PBS) or mFhSAP-2 emulsified in Montanide ISA50, which is a non-inflammatory and non-toxic adjuvant. Negative controls were immunized only with the adjuvant, and every single injection was applied with a sterile hypodermic G-26 needle connected to a 1-cc insulin syringe.
None of the control or immunized animals developed adverse reactions. The high variability you are referring to was observed exclusively in the group immunized with mFhSAP-2 without adjuvant. In this group, two of the five animals developed lower IgG titers, as well as lower IgG2a and IgG2c subtype levels than the other three animals. The most likely interpretation of this result is that mFhSAP-2 alone is not homogenously immunogenic in all animals. This is a common finding when small molecules are used as vaccines. In such cases, the use of an adjuvant is often convenient to enhance and homogenize the immune response, as was observed with the mFhSAP-2+ISA50 formulation. To adress this issue, section 3.5 of the revised manuscript now includes the individual values for the IgG2a and IgG2c antibody response for each animal in the mFhSAP-2 group, and an appropriate explanation with references has been added to the Discussion section.
- Line 160. Was the FBS heat inactivated or not? Where were the FBS and two antibiotics sourced from?
Response. Yes, FBS was heat inactivated. This detail, along with the source of FBS and the two antibiotics, were included in the revised version of the manuscript.
- Line 171. I assume this incubation was the overnight incubation in step 2.4? If so, clarify in the text.
Response. Yes, in section 2.4 the incubation was overnight, and this was informed in the text.
- Line 181. As stated above, what licence number was used to authorize this study and what adverse events were observed?
Response. Lines 181 to 197 correspond to an in vitro experiment with RAW264.7 cells for which no specific license is required. If this comment refers to the rabbits and mice infection sera used in the study, please see the IACUC approved protocol numbers (Protocols no. 7870104, and 7870106) mentioned in section 2.3. This is also referenced in section 2.7, in which is highlighted that these samples were obtained from the repository bank of the Molecular Parasitology and Immunology Laboratory, at the Department of Microbiology of the University of Puerto Rico-Medical Sciences Campus.
- Line 183 In what matrix were the parasites administered orally?
Response. Line 183 corresponds to an in vitro experiment with RAW264.7 cells. If you are referring to section 2.7, the metacercariae were administered orally suspended in tap water. As stated above, the infection sera from these animals were obtained from the repository bank of the Molecular Parasitology and Immunology Laboratory, at the Department of Microbiology of the University of Puerto Rico-Medical Sciences Campus. No challenge infections were performed as part of this study.
- Lines 186-190. It should be clearly stated whether or not rabbits and mice completely cleared or partially cleared infection. This is currently not clear. Were there any adverse effects seen? Did the animals loose weight or develop signs of infection? Please clarify in detail. Did this differ between mouse strains? This is critical to the hypothesis of the paper.
Response. Lines 186-190 correspond to an in vitro experiment with RAW264.7 cells. As stated in section 2.7, we did not perform any challenge infection in the present study. Therefore, questions regarding infection, weight loss, or adverse effects do not apply. We are unsure of the relevance of this comment to our study and our hypothesis, as these aspects were not part of our experimental design.
In section 2.7, we stated that the sera from F. hepatica infected animals (NZW rabbits and C57BL/6 mice) were obtained from the repository bank of the Molecular Parasitology and Immunology Laboratory at the Department of Microbiology of the University of Puerto Rico-Medical Sciences Campus. All animals had been orally infected with F. hepatica mc and all of them had developed active infection, which was confirmed by finding immature or mature flukes in the livers. These sera were used in our study with two purposes: 1) determining whether mFhSAP-2 retained its capacity to react with infection sera, and 2) comparing the IgG isotyping profile elicited by F. hepatica infection with those elicited by mFhSAP-2 immunization.
We demonstrated that, regardless the infection sera is from rabbits or mice, there are no differences in the antibody response associated with the animal species. Moreover, there were no differences between the antibody response elicited by infection compared to those induced by immunization with mFhSAP-2. However, immunization with mFhSAP-2+ISA50 elicited a significantly higher antibody response compared to immunization with mFhSAP-2 alone and F. hepatica infection. Please refer to section 3.4 and Figure 5 for further details.
- Line 194. Did sterile abscesses appear? Why was the dose administered in the same location? Were the animals restrained by the scruff during handling? Were there any adverse events associated with immunisation and if so, was it in all mice?
Response. Line 194 is about the measurement of TNF levels in RAW 264.7 cell supernatants. Regarding the immunization protocol, all subcutaneous injections were administered in compliance with the IACUC guidelines. Our personnel have been trained by an experienced veterinarian, familiar with the IACUC regulations and guidelines. All injections were administered at different sites on the dorsal surface of the mouse. The needle was inserted at a 45-to-90-degree angle to the pinched skin to ensure the needle was fully covered. One group was immunized with mFhSAP-2 alone, another with mFhSAP-2 + ISA50, while the control group received the adjuvant only. As mentioned above, the adjuvant Montanide ISA50 is not toxic, and as expected, there were no adverse events in any of the animals, and no sterile abscesses appeared. For restraining, the mouse was gently held by the mouse’s tail using the dominant hand. Then, using the forefinger and thumb of the non-dominant hand, we tented the loose skin over the shoulder, positioning our fingers just behind the ears to limit the mouse’s head movements. This is a standard operating protocol in our laboratory for immunizing animals via the subcutaneous route, without any change in the results associated with animal handling.
- Line 201. Was the sera stored for more than three hours? At what temperature were they stored? Were the sera left to clot at 37 degree C or put into SST tubes? Please add all of these details.
Response. Line 201 refers to the sera infection donated for the study (section 2.6). In section 2.7 of the revised manuscript, we have added more detailed information regarding the collection of sera from mFhSAP-2 and mFhSAP-2+ISA50 immunized animals.
- Line 222. Was the 30% peroxide solution a volume to volume dilution in water of a stock solution? If so, please state molarity of stock solution. Express concentration as molarity, not % v/v.
Response. The peroxide solution described in line 248 is commercially available as a stock solution at 30% weight in water (w/w). This information was added to the text in section 2.9.
- Line 224. How long were the plates left to incubate for?
Response. The plates were incubated with the substrate for 30 minutes at room temperature in the dark to allow the enzymatic reaction to develop. This information was added to the text in section 2.9.
- Line 225. Express concentration as molarity
Response. We believe that you are referring to the substrate molarity on line 247. The molarity of this buffer (0.1 M citrate-phosphate) was provided in that line.
- Line 246 “et al” not in italics
Response. We believe you are referring to the use of “et al” on line 275. If so, it is now in italics.
- Line 356 to 358. This statement is not justified. These determinations were single point, not serial dilutions (which would have provided titers) so the outcome does not justify this conclusion.
Response. We are having difficulty identifying the specific statement you are referring to, as lines 356 to 358 in our manuscript correspond to the expression of mFhSAP-2 in E. coli (section 3.1). The sections discussing the antibody responses (sections 3.4 and 3.5) indeed describe single-point determinations, not titers. The only section where serial dilutions and IgG2a/IgG2c titers are reported are found in section 3.6. Nevertheless, we respectfully clarify that our statements in sections 3.4 and 3.5 are based on the average OD values obtained at a single dilution (1:100), as described in section 2.9 in Material and Methods. The comparisons reflect relative antibody levels, not endpoint titers.
- Figure 4B. Did these mice fully clear the pathogen? Were the source sera taken early or late?
Response. The mice shown in Figure 4B (now Fig.5B) were not challenged with F. hepatica after immunization, and the infected mice were not immunized with mFhSAP-2. Therefore, there is no possibility that any of these mice partially or fully cleared the pathogen. This figure shows the reactivity of mFhSAP-2 with sera from mice with patent F. hepatica infection, measured by ELISA. As stated in section 2.6, the sera from the infected mice were collected on days 15 to 21 after the challenge infection, when the parasites are still migrating through the liver parenchyma or are close to enter the bile ducts.
- Line 395 does not currently mention IgG2A. Please add missing information
Response. We assumed you are referring to the line 442, if so, IgG2a was added to the text.
- Section 3.5 states that the outputs from these analyses exhibited large variances. I am concerned that the statistical analysis of such heteroscedastic data is not sound.
Response. This was already responded in question-17 and before. The main text was modified as stated above.
- Line 403. The variance of this data is very high indeed. Were there differences at the injection sites seen in these animals? Could abscesses account for the variance?
Response. As we have mentioned above, the variability occurred exclusively in the group immunized with mFhSAP-2 alone. Two of five animals (40%) developed notably lower IgG2a and IgG2c antibody levels than the other three animals. These two animals had average OD492 values for IgG2a of 0.461 and 0.756, whereas the other three animals had average OD492 values of 1.511, 1.516 and 1.754, respectively. Similarly, the same two animals had notably lower IgG2c levels (OD492 = 0.202 and 0.801) than the other three animals, which had average OD492 values of 3.08, 3.153 to 3.37, respectively. This variability was not related to differences in injection site or technique, as all animals immunized with mFhSAP-2 or mFhSAP-2+ISA50 were injected at the dorsal surface following the same standard protocol. No abscesses or adverse effects associated to the immunizations were observed in the animals. In the revised version of the paper, this was mentioned in section 3.5 of Results.
- Figure 5 A, B and C are currently bar charts. Section 3.5 clearly states that these data are heteroscedastic. Please change these graphs to scatterplot to reveal this variance in data. Are the multiple cross-tests sound? Do they assume homoscedasity?
Response. This point was addressed in our response to the general comments. Our analysis assumed homoscedasticity. Figure-5 (now Figure 6) was changed to a scatterplot to better illustrate the source of variance in the data.
- Figures 5 A, B and C. What are the cut-offs for each graph? They are not likely to be zero.
Response. The cut-off for each graph was included.
- Line 454. Delete extra space in title
Response. We believe you are referring to the extra space in the title on line 508, if so, it was deleted.
- Line 457 states “non-stimulated, immunized mice” then on line 458 “nonstimulated”. This sentence does not make sense to me.
Response. We believe you are referring to the splenocyte proliferation experiments, if so, this section (section 3.7), was revised and rewritten
- Line 460. What is the meaning of stimulated? Do mean immunized? If so, clarify. If not, also clarify
Response. In section 3.7, “stimulated” is not synonymous with “immunized”. This section refers to an in vitro proliferation assay and a flow cytometry assay to determine T-cell activation status. Splenocytes from negative controls and immunized animals were stimulated ex-vivo with mFhSAP-2 or left unstimulated. This distinction was clarified in the revised version. For further details, please refer to sections 2.11 and 2.12 in the Materials and Methods section.
- Line 465 Italics missing
- Line 466 – clarify meaning
- Line 468 Italics missing
- Line 523, delete comma
- Line 524, add “s” to suggest.
- Line 534 Correction required second word
- Line 542 Italics missing
- Line 544 Repetition of words – sentence does not make sense
Response. All these mistakes were localized and corrected.
- References: Many references have italics missing for species names
Response. All references were revised and italics in the species names were placed.
Reviewer 2 Report
Comments and Suggestions for Authors
The present manuscript (ID: vaccines-3545296) titled "A modified variant of Fasciola hepatica mFhSAP-2 as a recombinant vaccine candidate induces high-avidity IgG2c antibodies and enhances T cell activation in immunized C57BL/6 mice" presents a scientifically strong and well-structured study on a modified Fasciola hepatica vaccine candidate, demonstrating clear immunogenicity and Th1-biased immune responses in mice. The study presents a modified variant of FhSAP-2, aiming to improve solubility and purification while retaining immunogenicity. This modification is valuable because it addresses a known challenge in producing Fasciola hepatica vaccines. However, several weaknesses reduce its practical impact:
Abstract. It provides a good overview of what the study is about, including what was done, and the results. However, the authors need to briefly mention how this study could help in fighting infections in animals or humans.
Introduction. This section explains why the research is important and how it fits into the larger picture of fighting Fasciola hepatica infections.
The discussion about Th1 and Th2 immune responses is repeated multiple times across the section. It could be more concise by summarizing in one or two sentences.
The introduction mentions previous studies on FhSAP-2 but does not provide a comparison with other vaccine candidates. How does this compare to existing attempts?
The study focuses on mice, but there's no mention of whether this vaccine could be useful for human or just for veterinary use.
Methods. Lines 96-105: The protein purification steps are highly technical, discussing buffer compositions, enzyme treatments, and chromatography. A simplified explanation of how these steps help improve the vaccine would be useful. The authors should provide simplified sketch of the protein purification process, showing the major steps in an easy-to-understand flowchart.
Lines 120-124: The endotoxin removal process is mentioned, but the risk of residual endotoxins affecting results is not discussed. What measures were taken to ensure purity?
Results. Lines 362-365: The antibody responses in mice are impressive, but there's no direct comparison with other Fasciola hepatica vaccines. Did these results improve upon previous studies? Moreover, no vaccine safety data was provided.
Lines 375-380: While the study measures immune responses in mice, no discussion is made about how this would translate to livestock animals like cows or sheep, which are the actual hosts of Fasciola hepatica.
Discussion. This section did not address potential risks.
The study argues that the vaccine could work in larger animals, but side effects and potential risks (such as overactivation of the immune system) are not discussed.
Lines 580-591: The study confirms high-avidity antibodies, but no next steps are suggested. Will they test it on livestock? Will they improve the formulation further?
Conclusion. While it concludes that the vaccine is effective in mice, it does not suggest when or how it could move to livestock trials or field testing. Producing vaccines at commercial scale is difficult and expensive, yet no discussion is given on whether this vaccine would be cost-effective or easy to manufacture for widespread use.
Author Response
Response to the Reviewer # 2
The present manuscript (ID: vaccines-3545296) titled "A modified variant of Fasciola hepatica mFhSAP-2 as a recombinant vaccine candidate induces high-avidity IgG2c antibodies and enhances T cell activation in immunized C57BL/6 mice" presents a scientifically strong and well-structured study on a modified Fasciola hepatica vaccine candidate, demonstrating clear immunogenicity and Th1-biased immune responses in mice. The study presents a modified variant of FhSAP-2, aiming to improve solubility and purification while retaining immunogenicity. This modification is valuable because it addresses a known challenge in producing Fasciola hepatica vaccines. However, several weaknesses reduce its practical impact:
- It provides a good overview of what the study is about, including what was done, and the results. However, the authors need to briefly mention how this study could help in fighting infections in animals or humans.
Response. Authors greatly appreciate your opinion about our work, and we have made our best to improve it based on your recommendations. At the end of the abstract, a brief statement mentioning how our study could help in fighting infections in animals or humans was added. The paragraph added is the following:
“The present study highlighted the feasibility to induce Th1-associated immune responses in mice using mFhSAP-2 as an antigen. Further studies are required to assess the potential of the mFhSAP-2+ISA50 formulation as a vaccine against F. hepatica in natural hosts, such as cattle and sheep, which could contribute to improved control and aid in the prevention and eradication of F. hepatica infection.”.
- This section explains why the research is important and how it fits into the larger picture of fighting Fasciola hepatica infections.
The discussion about Th1 and Th2 immune responses is repeated multiple times across the section. It could be more concise by summarizing in one or two sentences.
The introduction mentions previous studies on FhSAP-2 but does not provide a comparison with other vaccine candidates. How does this compare to existing attempts?
The study focuses on mice, but there's no mention of whether this vaccine could be useful for human or just for veterinary use.
Response. Authors appreciated these recommendations and based on the extension of these recommendation we decided to rewrite the introduction. In the new version the discussion about Th1/Th2 immune response was significantly reduced. Vaccine candidates other than FhSAP-2 were mentioned and compared, as well as the potential use of FhSAP-2 as a vaccine candidate for human or veterinary use.
Methods. Lines 96-105: The protein purification steps are highly technical, discussing buffer compositions, enzyme treatments, and chromatography. A simplified explanation of how these steps help improve the vaccine would be useful. The authors should provide simplified sketch of the protein purification process, showing the major steps in an easy-to-understand flowchart.
Response. A simplified flowchart illustrating the main steps of the purification process was provided (Fig. 2) and a detailed description of the process including buffer composition, enzymes and chromatography was also provided as Supplementary data (DataS1).
Lines 120-124: The endotoxin removal process is mentioned, but the risk of residual endotoxins affecting results is not discussed. What measures were taken to ensure purity?
Response. The endotoxin removal is an exhaustive process in which the protein is suggested to successive passes through a PMB column. This process continue until the levels of endotoxin assessed by a Chromogenic Limulus Amebocyte Lysate assay are lower than 0.1 EU/ug (Endotoxin unit per microgram) or EU/ml depending on the sensitivity of the assay. In our case, the purified protein was recovered with endotoxin levels that were lower than 0.19 EU/mg, which indicate that only trace amounts of endotoxin remained in the protein. The purity was assessed by densitometric analysis of Commassie blue stained SDS-PAGE, with a purity of >85%. All these details were added to Material & Methods and Results sections.
Results. Lines 362-365: The antibody responses in mice are impressive, but there's no direct comparison with other Fasciola hepatica vaccines. Did these results improve upon previous studies? Moreover, no vaccine safety data was provided.
Response. In our opinion, no matter how robust the antibody response elicited against a determined antigen may appear, what is most important is the quality of this response. If a F. hepatica vaccine induces high IgG antibody titers and these antibodies are mostly IgG1-type, it is highly probable these antibodies will not be protective, as has been observed in natural F. hepatica infection. In contrast, if the antibody response induced by the vaccine is mostly IgG2(a/c)-type, then it is highly possible that the vaccine is protective. Therefore, the IgG1/IgG2 ratio and IL4-IFN ratio are among the most important criteria, as they are clear indicators of a trend toward a Th1 or Th2 immune response. However, this information is largely absent from most studies about Fasciola hepatica vaccines. This is particularly the case when vaccines are assessed in sheep, goat or cattle due to few or no commercially available kits for determining IgG subtypes or cytokine profile in these species.
In our study, we emphasized the need to elicit low IgG1/IgG2a or IgG1/IgG2c ratios as well as low IL-4/IFN ratios based on the studies of Pleasance et al. (2011) with ITT sheep, which exhibit natural resistance to the liver fluke infection. In the revised version of the manuscript these issues were added in both the Introduction and Discussion sections. Unfortunately, we have not yet safety data for this vaccine. This is the first study done with the modified FhSAP-2. Firstly, we must demonstrate that this construction induced a similar antibody/cytokine pattern to the full-length protein. Once this was confirmed, our next steps will be to perform studies aimed for determining the efficacy and safety of mFhSAP-2+ISA50 vaccine during the lifespan of mice (1 year). This will include examining the durability of the IgG2a and IgG2c titers, the avidity and affinity of these antibodies, as well as identify major adverse effect in the lungs, kidneys, or liver, while evaluating vaccine-induced immune protection. These statements were added to the Discussion in the revised version.
Lines 375-380: While the study measures immune responses in mice, no discussion is made about how this would translate to livestock animals like cows or sheep, which are the actual hosts of Fasciola hepatica.
Response. A statement about how the measurements of antibodies would translate to livestock was added to the revised manuscript. As mentioned above, it would be mandatory to first determine the optimized dose and frequency of injection that maximizes the levels of protection in sheep or cattle. Then, the measurements of IgG1/IgG2-antibodies and IL-4/IFN will serve to determine Th1 or Th2 polarization and thus, determine if the results observed in mice are translated to livestock. These statements were added to the Discussion in the revised version.
Discussion. This section did not address potential risks.
The study argues that the vaccine could work in larger animals, but side effects and potential risks (such as overactivation of the immune system) are not discussed.
Response. Potential risks of the vaccine were considered and added to the Discussion.
Lines 580-591: The study confirms high-avidity antibodies, but no next steps are suggested. Will they test it on livestock? Will they improve the formulation further?
Response. The avidity of antibodies elicited in livestock will be measured. Steps to improve the formulation if needed were also considered. These considerations were added to the Discussion in the revised version.
Conclusion. While it concludes that the vaccine is effective in mice, it does not suggest when or how it could move to livestock trials or field testing. Producing vaccines at commercial scale is difficult and expensive, yet no discussion is given on whether this vaccine would be cost-effective or easy to manufacture for widespread use.
Response. These aspects were considered and added to the Conclusions.

Reviewer 3 Report
Comments and Suggestions for Authors
The objective of this study is clearly articulated, the study focuses on the idea of testing a modified 9.8kDa variant of mFhSAP-2, lacking the string of 15 hydrophobic amino acids at amino terminus containing a dominant Th1-epitope, whether it retain its immunogenic and Th1-inducing properties.
The methodology is well-presented and effectively demonstrates the study’s design. The methodology is sufficient to test the proposed hypothesis.
The results and figures align well with the study plan outlined above. They are clearly presented and of sufficient quality.
The conclusion is well-supported by the findings. However, the discussion should be more focused, with greater emphasis on the other potential vaccine candidates. My recommendation is to highlight the advantages of modified FhSAP-2 over unmodified vaccine candidate, and also over other vaccines against Fasciola, particularly in relation to the findings of this study, to reinforce the study’s novelty and impact.
Additionally, the limitations of the study are not discussed. A dedicated part of the Discussion section addressing potential weaknesses would provide a more balanced and transparent evaluation of the study’s implications.
Author Response
Response to the Reviewer # 3
The objective of this study is clearly articulated, the study focuses on the idea of testing a modified 9.8kDa variant of mFhSAP-2, lacking the string of 15 hydrophobic amino acids at amino terminus containing a dominant Th1-epitope, whether it retain its immunogenic and Th1-inducing properties.
The methodology is well-presented and effectively demonstrates the study’s design. The methodology is sufficient to test the proposed hypothesis.
The results and figures align well with the study plan outlined above. They are clearly presented and of sufficient quality.
The conclusion is well-supported by the findings. However, the discussion should be more focused, with greater emphasis on the other potential vaccine candidates. My recommendation is to highlight the advantages of modified FhSAP-2 over unmodified vaccine candidate, and also over other vaccines against Fasciola, particularly in relation to the findings of this study, to reinforce the study’s novelty and impact.
Additionally, the limitations of the study are not discussed. A dedicated part of the Discussion section addressing potential weaknesses would provide a more balanced and transparent evaluation of the study’s implications.
Response. Authors greatly appreciate your opinion about our work. Your recommendations have been considered. We have improved the discussion addressing all these issues, making particular emphasis in the advantage of mFhSAP-2 over other vaccine candidates as well as addressing potential weaknesses and limitations of the study.
